# DIFFERENTIALLY PRIVATE ONE PERMUTATION HASHING

## ABSTRACT

Minwise hashing (MinHash) is a standard hashing algorithm for large-scale search and learning with the binary Jaccard similarity. One permutation hashing (OPH) is an effective and efficient alternative of MinHash which splits the data into $K$ bins and generates hash values within each bin. In this paper, to protect the privacy of the output sketches, we combine differential privacy (DP) with OPH, and propose DP-OPH framework with three variants: DP-OPH-fix, DP-OPH-re and DP-OPH-rand, depending on the densification strategy to deal with empty bins in OPH. A detailed roadmap to the algorithm design is presented along with the privacy analysis. Comparisons of our DP-OPH methods with the DP minwise hashing (DP-MH) alternative are provided to justify the advantage of DP-OPH. Experiments on similarity search confirm the effectiveness of our proposed algorithms, and provide guidance on the choice of proper variant in different scenarios. We also provide an extension to real-value data, named DP-BCWS, in the appendix.

## 1 INTRODUCTION

Let $\boldsymbol{u}, \boldsymbol{v} \in \{0, 1\}^D$ be two $D$-dimensional binary vectors. In this paper, we focus on the hashing algorithms for the Jaccard similarity (a.k.a. the "resemblance") defined as

$$J(\boldsymbol{u}, \boldsymbol{v}) = \frac{\sum_{i=1}^{D} \mathbb{1}\{\boldsymbol{u}_i = \boldsymbol{v}_i = 1\}}{\sum_{i=1}^{D} \mathbb{1}\{\boldsymbol{u}_i + \boldsymbol{v}_i \geq 1\}}. \tag{1}$$

This is a widely used similarity measure in machine learning applications. $\boldsymbol{u}$ and $\boldsymbol{v}$ can also be viewed as two sets of items represented by the locations of non-zero entries. In industrial applications with massive data size, directly calculating the pairwise Jaccard similarity among the data points becomes too expensive. To accelerate large-scale search and learning, the celebrated *"minwise hashing"* (MinHash) algorithm (Broder, 1997; Broder et al., 1997) has been a standard hashing technique for approximating the Jaccard similarity in massive binary datasets. It has seen numerous applications such as near neighbor search, duplicate detection, malware detection, clustering, large-scale learning, social networks, and computer vision (Indyk & Motwani, 1998; Charikar, 2002; Fetterly et al., 2003; Das et al., 2007; Buehrer & Chellapilla, 2008; Bendersky & Croft, 2009; Chierichetti et al., 2009; Pandey et al., 2009; Lee et al., 2010; Deng et al., 2012; Chum & Matas, 2012; Tamersoy et al., 2014; Shrivastava & Li, 2014; Zhu et al., 2017; Nargesian et al., 2018; Wang et al., 2019; Lemiesz, 2021; Feng & Deng, 2021; Li & Li, 2022). The output of MinHash is an integer. For large-scale applications, to store and use the hash values (or called sketches) more conveniently and efficiently, Li & König (2010) proposed $b$-bit MinHash that only stores the last $b$ bits of the hashed integers, which is memory-efficient and convenient for similarity search and machine learning. Thus, it has been a popular coding strategy for the MinHash values and its alternatives (Li et al., 2011; 2015; Shah & Meinshausen, 2017; Yu & Weber, 2022).

### 1.1 ONE PERMUTATION HASHING (OPH) FOR JACCARD SIMILARITY

To use MinHash in practice, we need to generate $K$ hash values to achieve good utility. This requires applying $K$ random permutations (or hash functions as approximations) per data point, yielding an $O(Kf)$ complexity where $f$ is the number of non-empty entries of the data. The method of one permutation hashing (OPH) (Li et al., 2012) provides a promising way to significantly reduce the complexity to $O(f)$. The idea of OPH is: to generate $K$ hashes, we split the data vector into $K$ non-overlapping bins, and conduct MinHash within each bin. Yet, empty bins may arise which breaks the alignment of the hashes such that the hash values do not form a metric space. To deal with empty bins, densification schemes (Shrivastava, 2017; Li et al., 2019) are proposed that fill the empty bins

with some non-empty bin. It is shown that OPH with densification also provides unbiased Jaccard estimator, and the estimation variance can often be smaller than that of MinHash. OPH has been widely used as an improved method over MinHash for the Jaccard similarity (Dahlgaard et al., 2017; Zhao et al., 2020; Jia et al., 2021; Tseng et al., 2021; Jiang et al., 2022).

## 1.2 HASHING/SKETCHING AND DIFFERENTIAL PRIVACY

At a higher level, MinHash and OPH both belong to the broad family of hashing/sketching methods, which generate sketches for data samples that are designed for various purposes and tasks. Examples of more sketching methods include the random projection (RP) based methods for cosine preserving (Charikar, 2002; Vempala, 2005), the count-sketch (CS) for frequency estimation (Charikar et al., 2004), and the Flajolet-Martin (FM) sketch (Flajolet & Martin, 1985) and HyperLogLog sketch (Flajolet et al., 2007) for cardinality estimation, etc. Since the data sketches produce "summaries" of the data which contain the original data information, sketching/hashing may also cause data privacy leakage. Therefore, protecting the privacy of the data sketches becomes an important topic which has gained growing research interests in recent years.

Differential privacy (DP) (Dwork et al., 2006b) has become a popular privacy definition with rigorous mathematical formulation, which has been widely applied to clustering, regression and classification, principle component analysis, matrix completion, optimization, deep learning (Blum et al., 2005; Chaudhuri & Monteleoni, 2008; Feldman et al., 2009; Gupta et al., 2010; Chaudhuri et al., 2011; Kasiviswanathan et al., 2013; Zhang et al., 2012; Abadi et al., 2016; Agarwal et al., 2018; Ge et al., 2018; Wei et al., 2020; Dong et al., 2022), etc. Prior efforts have also been conducted to combine differential privacy with the aforementioned hashing algorithms (e.g.,for RP (Blocki et al., 2012; Kenthapadi et al., 2013; Stausholm, 2021), count-sketch (Zhao et al., 2022), and FM sketch (Smith et al., 2020; Dickens et al., 2022)). Some works (e.g., Blocki et al. (2012); Smith et al. (2020); Dickens et al. (2022)) assumed "internal randomness", i.e., the randomness of the hash functions are kept private, and showed that many hashing methods themselves already possess strong DP property under some data conditions. However, this setting is more restrictive in practice as it requires the hash keys or projection matrices cannot be accessed by any adversary. In another setup (e.g., Kenthapadi et al. (2013); Stausholm (2021); Zhao et al. (2022)), both the randomness of the hash functions and the algorithm outputs are treated as public information, and perturbation mechanisms are developed to make the algorithms differentially private.

## 1.3 OUR CONTRIBUTIONS

While prior works have proposed DP algorithms for some sketching methods mentioned earlier, the differential privacy of OPH and MinHash for the Jaccard similarity has not been well studied. In this paper, we mainly focus on the differential privacy of one permutation hashing (OPH), the state-of-the-art framework for hashing the Jaccard similarity. We consider the more practical and general setup where the randomness of the algorithm is "external" and public.

We develop three variants under the DP-OPH framework, DP-OPH-fix, DP-OPH-re, and DP-OPH-rand, corresponding to fixed densification, re-randomized densification, and no densification for OPH, respectively. We provide detailed algorithm design and privacy analysis for each variant, and compare them with a DP MinHash (DP-MH) method. In our retrieval experiments, we show that the proposed DP-OPH method substantially improves DP-MH, and re-randomized densification is superior over fixed densification in terms of differential privacy. DP-OPH-rand performs the best when $\epsilon$ is small, while DP-OPH-re is the most performant in when larger $\epsilon$ is allowed.

## 2 BACKGROUND: MINHASH, $b$-BIT CODING, AND DIFFERENTIAL PRIVACY

---

**Algorithm 1** Minwise hashing (MinHash)

---

**Input:** Binary vector $\boldsymbol{u} \in \{0, 1\}^D$; number of hash values $K$
**Output:** $K$ MinHash values $h_1(\boldsymbol{u}), ..., h_K(\boldsymbol{u})$
  1: Generate $K$ independent permutations $\pi_1, ..., \pi_K \colon [D] \to [D]$ with seeds $1, ..., K$ respectively
  2: **for** $k = 1$ to $K$ **do**
  3:     $h_k(\boldsymbol{u}) \leftarrow \min_{i:u_i \neq 0} \pi_k(i)$
  4: **end for**

---

**Minwise hashing (MinHash).** The MinHash method is summarized in Algorithm 1. We first generate $K$ independent permutations $\pi_1, ..., \pi_K : [D] \mapsto [D]$, where the seeds ensure that all data vectors use the same set of permutations. Here, $[D]$ denotes $\{1, ..., D\}$. For each permutation, the hash value is simply the first non-zero location in the permuted vector, i.e., $h_k(\boldsymbol{u}) = \min_{i:v_i \neq 0} \pi_k(i)$, $\forall k = 1, ..., K$. Analogously, for another data vector $\boldsymbol{v} \in \{0,1\}^D$, we also obtain $K$ hash values, $h_k(\boldsymbol{v})$. The MinHash estimator of $J(\boldsymbol{u}, \boldsymbol{v})$ is the average over the hash collisions:

$$\hat{J}_{MH}(\boldsymbol{u}, \boldsymbol{v}) = \frac{1}{K} \sum_{k=1}^{K} \mathbb{1}\{h_k(\boldsymbol{u}) = h_k(\boldsymbol{v})\}, \tag{2}$$

where $\mathbb{1}\{\cdot\}$ is the indicator function. By some standard probability calculations, we can show that

$$\mathbb{E}[\hat{J}_{MH}] = J, \qquad Var[\hat{J}_{MH}] = \frac{J(1-J)}{K}.$$

In practice, $K$ does not need to be very large to achieve good utility. For instance, usually $128 \sim 1024$ hash values would be sufficient for search and learning problems (Indyk & Motwani, 1998; Li et al., 2011; Shrivastava & Li, 2014).

$b$**-bit coding of the hash value.** Li & König (2010) proposed "$b$-bit minwise hashing" as a convenient coding strategy for the integer hash value $h(\boldsymbol{u})$ generated by MinHash (or by OPH which will be introduced later). Basically, we only keep the last $b$-bits of each hash value. In our analysis, for convenience, we assume that "taking the last $b$-bits" can be achieved by some "rehashing" trick to map the integer values onto $\{0, ..., 2^b - 1\}$ uniformly. There are at least three benefits of this coding strategy: (i) storing only $b$ bits saves the storage cost compared with storing the full 32 or 64 bit integers; (ii) the last few bits are more convenient for the purpose of indexing, e.g., in approximate nearest neighbor search (Indyk & Motwani, 1998); (iii) we can transform the last few bits into a positional representation, allowing us to approximate the Jaccard similarity by inner product, which is required by training large-scale linear models (Li et al., 2011). Given these advantages, in this work, we will adopt this $b$-bit coding strategy in our private algorithm design.

**Differential privacy (DP).** We formally define differential privacy (DP) as follows.

**Definition 2.1** (Differential privacy (Dwork et al., 2006b)). *For a randomized algorithm $\mathcal{M} : \mathcal{U} \mapsto Range(\mathcal{M})$ and $\epsilon, \delta \geq 0$, if for any two neighboring datasets $U$ and $U'$, it holds that*

$$Pr[\mathcal{M}(U) \in Z] \leq e^\epsilon Pr[\mathcal{M}(U') \in Z] + \delta$$

*for $\forall Z \subset Range(\mathcal{M})$, then algorithm $\mathcal{M}$ is said to satisfy $(\epsilon, \delta)$-differentially privacy. If $\delta = 0$, $\mathcal{M}$ is called $\epsilon$-differentially private.*

Intuitively, DP requires the distributions of the outputs before and after a small change in the data are similar so that an adversary cannot detect the change based on the outputs. Smaller $\epsilon$ and $\delta$ implies stronger privacy. The parameter $\delta$ is usually interpreted as the "failure probability" allowed for the $\epsilon$-DP guarantee to be violated. In our work, we follow the standard definition in aforementioned related works on DP hashing: $\boldsymbol{u}, \boldsymbol{u}' \in \{0,1\}^D$ are called neighboring if they differ in one dimension.

**Privacy statement and applications.** The above definition of adjacency leads to the "attribute-level" DP. Treating the binary vectors as sets, with our proposed DP-OPH algorithms, *an adversary cannot detect from the output sketches whether any item exists in the set or not, which holds independently for all the data vectors in the database*. As a concrete example application, the bioinformatics community releases sets of 1000 MinHashes for all known genomes on a regular basis (Ondov et al., 2016; Brown & Irber, 2016), which are used for various ML tasks like classification, clustering, etc. In this type of data, each data point corresponds to (a large set of) genes of a human, which contains the basic biological information of an individual. Hence, it is highly sensitive and confidential. Our methods protect the identification of any gene from the DP-OPH sketches in the DP sense.

## 3 HASHING FOR JACCARD SIMILARITY WITH DIFFERENTIAL PRIVACY

As discussed earlier, one permutation hashing (OPH) (Li et al., 2012) is a popular and highly efficient hashing algorithm for the Jaccard similarity. In this section, we present our main algorithms called DP-OPH based on privatizing the $b$-bit hash values from OPH. In addition, we compare it with a differentially private MinHash alternative named DP-MH.

---

**Algorithm 2** One Permutation Hashing (OPH)

---

**Input:** Binary vector $\boldsymbol{u} \in \{0,1\}^D$; number of hash values $K$
**Output:** $K$ OPH hash values $h_1(\boldsymbol{u}), ..., h_K(\boldsymbol{u})$

1: Let $d = D/K$. Use a permutation $\pi : [D] \mapsto [D]$ with fixed seed to randomly split $[D]$ into $K$ equal-size bins $\mathcal{B}_1, ..., \mathcal{B}_K$, with $\mathcal{B}_k = \{j \in [D] : (k-1)d + 1 \leq \pi(j) \leq kd\}$
2: **for** $k = 1$ to $K$ **do**
3:      **if** Bin $\mathcal{B}_k$ is non-empty **then**
4:          $h_k(\boldsymbol{u}) \leftarrow \min_{j \in \mathcal{B}_k, u_j \neq 0} \pi(j)$
5:      **else**
6:          $h_k(\boldsymbol{u}) \leftarrow E$
7:      **end if**
8: **end for**

---

**Algorithm 3** Densification for OPH, two options: fixed and re-randomized

---

**Input:** OPH hash values $h_1(\boldsymbol{u}), ..., h_K(\boldsymbol{u})$ each in $[D] \cup \{E\}$; bins $\mathcal{B}_1, ..., \mathcal{B}_K$; $d = D/K$
**Output:** $K$ densified OPH hash values $h_1(\boldsymbol{u}), ..., h_K(\boldsymbol{u})$

1: Let $NonEmptyBin = \{k \in [K] : h_k(\boldsymbol{u}) \neq E\}$
2: **for** $k = 1$ to $K$ **do**
3:      **if** $h_k(\boldsymbol{u}) = E$ **then**
4:          Uniformly randomly select $k' \in NonEmptyBin$
5:          $h_k(\boldsymbol{u}) \leftarrow h_{k'}(\boldsymbol{u})$                   ▷ OPH-fix: fixed densification
6:          **Or**
7:          $MapToIndex = SortedIndex\left(\pi(\mathcal{B}_k)\right) + (k'-1)d$
8:          $\pi^{(k)} : \pi(\mathcal{B}_{k'}) \mapsto MapToIndex$        ▷ within-bin partial permutation
9:          $h_k(\boldsymbol{u}) \leftarrow \min_{j \in \mathcal{B}_{k'}, u_j \neq 0} \pi^{(k)}\left(\pi(j)\right)$     ▷ OPH-re: re-randomized densification
10:      **end if**
11: **end for**

---

## 3.1 ONE PERMUTATION HASHING (OPH)

As outlined in Algorithm 2, the procedure of OPH is simple: we first use a permutation $\pi$ (same for all data vectors) to randomly split the feature dimensions $[D]$ into $K$ bins $\mathcal{B}_1, ..., \mathcal{B}_K$ with equal length $d = D/K$ (assuming integer division holds). Then, for each bin $\mathcal{B}_k$, we set the smallest permuted index of "1" as the $k$-th OPH hash value. If $\mathcal{B}_k$ is empty (i.e., it does not contain any "1"), we record an "$E$" representing empty bin. Li et al. (2012) showed that we can construct statistically unbiased Jaccard estimators by ignoring the empty bins. However, this estimator is unstable when the data is relatively sparse; moreover, since empty bins are different for every distinct data vector, the vanilla OPH hash values do not form a metric space (i.e., do not satisfy the triangle inequality).

**Densification for OPH.** To tackle the issue caused by empty bins, a series of works has been conducted to densify the OPH. The general idea is to "borrow" the data/hash from non-empty bins, with some careful design. In Algorithm 3, we present two recent representatives of OPH densification methods: fixed densification (Shrivastava, 2017) and re-randomized densification (Li et al., 2019), noted as OPH-fix and OPH-re, respectively. Given an OPH hash vector from Algorithm 2 (possibly containing "$E$"s), we denote the set of non-empty bins $NonEmptyBin = \{k \in [K] : h_k(\boldsymbol{u}) \neq E\}$. The densification procedure scans over $k = 1, ..., K$. For each $k$ with $h_k(\boldsymbol{u}) = E$, we do:

1. Uniformly randomly pick a bin $k' \in NonEmptyBin$ that is non-empty.
2. (a) OPH-fix: we directly copy the $k'$-th hash value: $h_k(\boldsymbol{u}) \leftarrow h_{k'}(\boldsymbol{u})$.
   (b) OPH-re: we apply an additional minwise hashing to bin $\mathcal{B}_{k'}$ using the "partial permutation" of $\mathcal{B}_k$ to get the hash for $h_k(\boldsymbol{u})$.

More precisely, for re-randomized densification, in Algorithm 3, $MapToIndex$ defines the "partial permutation" of bin $\mathcal{B}_k$, where the function $SortedIndex$ returns the original index of an sorted array. For example, let $D = 16$, $K = 4$, and $d = D/K = 4$ and suppose the indices in each bin are in ascending order, and $\mathcal{B}_2 = [1, 5, 13, 15]$ is empty. Suppose $\pi(13) = 5, \pi(5) = 6, \pi(1) = 7, \pi(15) = 8$. In this case, $\pi(\mathcal{B}_2) = [7, 6, 5, 8]$, so $SortedIndex(\pi(\mathcal{B}_2)) = [3, 2, 1, 4]$. Assume $k' = 3$ is picked and $\pi(\mathcal{B}_3) = [9, 12, 10, 11]$. At line 7 we have $MapToIndex = [11, 10, 9, 12]$

---

**Algorithm 4** Differentially Private Densified One Permutation Hashing (DP-OPH-fix, DP-OPH-re)

**Input:** Densified OPH hash values $h_1(\boldsymbol{u}), ..., h_K(\boldsymbol{u})$; number of bits $b$; $\epsilon > 0, 0 < \delta < 1$
$\quad\quad f$: lower bound on the number of non-zeros in each data vector
**Output:** $b$-bit DP-OPH values $\tilde{h}(\boldsymbol{u}) = [\tilde{h}_1(\boldsymbol{u}), ..., \tilde{h}_K(\boldsymbol{u})]$
1: Take the last $b$ bits of all hash values $\quad\quad\quad\quad\quad\quad\quad\quad$ ▷ After which $h_k(\boldsymbol{u}) \in \{0, ..., 2^b - 1\}$
2: Set $N = F_{fix}^{-1}(1 - \delta; D, K, f)$ (for DP-OPH-fix) or $N = F_{re}^{-1}(1 - \delta; D, K, f)$ (for DP-OPH-re),
$\quad$ and $\epsilon' = \epsilon/N$
3: **for** $k = 1$ to $K$ **do**
4: $\quad \tilde{h}_k(\boldsymbol{u}) = \begin{cases} h_k(\boldsymbol{u}), & \text{with probability } \frac{e^{\epsilon'}}{e^{\epsilon'} + 2^b - 1} \\ i, & \text{with probability } \frac{1}{e^{\epsilon'} + 2^b - 1}, \text{ for } i \in \{0, ..., 2^b - 1\}, \ i \neq h_k(\boldsymbol{u}) \end{cases}$
5: **end for**

---

and at line 8, $\pi^{(2)}$ is a mapping $[9, 12, 10, 11] \mapsto [11, 10, 9, 12]$, which effectively defines another within-bin permutation of $\pi(\mathcal{B}_3)$ using the partial ordering of $\pi(\mathcal{B}_2)$. Finally, we set $h_k(\boldsymbol{u})$ as the minimal index of "1" among the additionally permuted elements in bin $\mathcal{B}_{k'}$.

We remark that in step 1, for any empty bin $k$, the "sequence" for non-empty bin lookup should be the same for any data vector. In practice, this can be achieved by simply seeding a random permutation of $[K]$ for each $k$. For instance, for $k = 1$ (when the first bin is empty), we always search in the order $[3, 1, 2, 4]$ until one non-empty bin is found, for all the data vectors.

It is shown that for both variants, the Jaccard estimator of the same form as (2) is unbiased. Li et al. (2019) showed that re-randomized densification always achieves smaller Jaccard estimation variance than that of fixed densification, and the improvement is especially significant when the data is sparse (see the reference paper for more details). Similar to $b$-bit minwise hashing, we can also keep the last $b$ bits of the OPH hash values to use them conveniently in search and learning.

### 3.2 DIFFERENTIAL PRIVATE ONE PERMUTATION HASHING (DP-OPH)

**DP-OPH with densification.** To privatize densified OPH, in Algorithm 4, we first take the last $b$ bits of the hash values. Since the output space is finite with cardinality $2^b$, we apply the randomized response technique (Dwork & Roth, 2014; Wang et al., 2017) to flip the bits to achieve DP. After running Algorithm 3, suppose a densified OPH hash value $h_k(\boldsymbol{u}) = j$, $j \in 0, ..., 2^b - 1$. With some $\epsilon' > 0$ that will be specified later, we output $\tilde{h}_k(\boldsymbol{u}) = j$ with probability $\frac{e^{\epsilon'}}{e^{\epsilon'} + 2^b - 1}$, and $\tilde{h}_k(\boldsymbol{u}) = i$ for $i \neq j$ with probability $\frac{1}{e^{\epsilon'} + 2^b - 1}$. It is easy to verify that, for a neighboring data $\boldsymbol{u}'$, when $h_k(\boldsymbol{u}') = j$, for $\forall i \in 0, ..., 2^b - 1$, we have $\frac{P(\tilde{h}_k(\boldsymbol{u}) = i)}{P(\tilde{h}_k(\boldsymbol{u}') = i)} = 1$; when $h_k(\boldsymbol{u}') \neq j$, we have $e^{-\epsilon'} \leq \frac{P(\tilde{h}_k(\boldsymbol{u}) = i)}{P(\tilde{h}_k(\boldsymbol{u}') = i)} \leq e^{\epsilon'}$. Therefore, for a single hash value, this bit flipping satisfies $\epsilon'$-DP.

It remains to determine $\epsilon'$. Naively, since the perturbations (flipping) of the hash values are independent, by the composition property of DP (Dwork et al., 2006a), simply setting $\epsilon' = \epsilon/K$ for all $K$ MinHash values would achieve overall $\epsilon$-DP (for the hashed vector). However, since $K$ is usually around hundreds, a very large $\epsilon$ value is required for this strategy to be useful. To this end, we can trade a small $\delta$ in the DP definition for a significantly reduced $\epsilon$. Note that, not all the $K$ hashed bits will change after we switch from $\boldsymbol{u}$ to its neighbor $\boldsymbol{u}'$. Assume each data vector contains at least $f$ non-zeros, which is realistic since many data in practice have both high dimensionality $D$ as well as many non-zero elements. Intuitively, when the data is not too sparse, $\boldsymbol{u}$ and $\boldsymbol{u}'$ tends to be similar (since they only differ in one element). Thus, the number of different hash values from Algorithm 3, $X = \sum_{k=1}^{K} \mathbb{1}\{h_k(\boldsymbol{u}) \neq h_k(\boldsymbol{u}')\}$, can be upper bounded by some $N$ with high probability $1 - \delta$. In the proof, this allows us to set $\epsilon' = \epsilon/N$ in the flipping probabilities and count $\delta$ as the failure probability in $(\epsilon, \delta)$-DP. In Lemma 3.1, we derive the precise probability distribution of $X$. Based on this result, in Algorithm 4, we set $N = F_{fix}^{-1}(1 - \delta; D, f, K)$ for DP-OPH-fix, $N = F_{re}^{-1}(1 - \delta; D, f, K)$ for DP-OPH-re, where $F_{fix}(x) = P(X \leq x)$ is the cumulative mass function (CMF) of $X$ with OPH-fix ((3) + (4)), and $F_{re}$ is the cumulative mass function of $X$ with OPH-re ((3) + (5)), and $F^{-1}$ is the inverse CMF. The distribution of $X$ is given as below. The proof can be found in Appendix C.

---

**Algorithm 5** Differentially Private One Permutation Hashing with Random Bits (DP-OPH-rand)

---

**Input:** OPH hash values $h_1(\boldsymbol{u}), ..., h_K(\boldsymbol{u})$ from Algorithm 2; number of bits $b$; $\epsilon > 0$
**Output:** DP-OPH-rand hash values $\tilde{h}(\boldsymbol{u}) = [\tilde{h}_1(\boldsymbol{u}), ..., \tilde{h}_K(\boldsymbol{u})]$
1: Take the last $b$ bits of all hash values                    $\triangleright$ After which $h_k(\boldsymbol{u}) \in \{0, ..., 2^b - 1\}$
2: **for** $k = 1$ to $K$ **do**
3:     **if** $h_k(\boldsymbol{u}) \neq E$ **then**
4:         $\tilde{h}_k(\boldsymbol{u}) = \begin{cases} h_k(\boldsymbol{u}), & \text{with probability } \frac{e^\epsilon}{e^\epsilon + 2^b - 1} \\ i, & \text{with probability } \frac{1}{e^{\epsilon'} + 2^b - 1}, \text{ for } i \in \{0, ..., 2^b - 1\}, \ i \neq h_k(\boldsymbol{u}) \end{cases}$
5:     **else**
6:         $\tilde{h}_k(\boldsymbol{u}) = i$ with probability $\frac{1}{2^b}$, for $i = 0, ..., 2^b - 1$  $\triangleright$ Assign random bits to empty bin
7:     **end if**
8: **end for**

---

**Lemma 3.1.** *Consider* $\boldsymbol{u} \in \{0,1\}^D$, *and denote* $f = |\{i : u_i = 1\}|$. *Let* $\boldsymbol{u}'$ *be a neighbor of* $\boldsymbol{u}$. *Denote* $X = \sum_{k=1}^K \mathbb{1}\{h_k(\boldsymbol{u}) \neq h_k(\boldsymbol{u}')\}$ *where the hash values are generated by Algorithm 3.* *Denote* $d = D/K$. *We have, for* $x = 0, ..., K - \lceil f/d \rceil$,

$$P(X = x) = \sum_{j=\max(0, K-f)}^{K - \lceil f/d \rceil} \sum_{z=1}^{\min(f,d)} \tilde{P}(x|z,j) P\left(\tilde{f} = z | K - j\right) P(N_{emp} = j), \quad (3)$$

*where* $P\left(\tilde{f} = z | K - j\right)$ *is given in Lemma C.2 and* $P(N_{emp} = j)$ *is from Lemma C.1. Moreover,*

**For OPH-fix:** $\tilde{P}(x|z,j) = \mathbb{1}\{x = 0\}\left(1 - P_{\neq}\right) + \mathbb{1}\{x > 0\}P_{\neq} \cdot g_{bino}\left(x - 1; \frac{1}{K-j}, j\right),$ (4)

**For OPH-re:** $\tilde{P}(x|z,j) = \left(1 - P_{\neq}\right) \cdot g_{bino}\left(x; \frac{P_{\neq}}{K-j}, j\right) + P_{\neq} \cdot g_{bino}\left(x - 1; \frac{P_{\neq}}{K-j}, j\right),$ (5)

*where* $g_{bino}(x; p, n)$ *is the probability mass function of* $Binomial(p, n)$ *with* $n$ *trials and success rate* $p$, *and* $P_{\neq}(z, b) = \left(1 - \frac{1}{2^b}\right)\frac{1}{z}$.

The privacy guarantee of DP-OPH with densification is shown as below.

**Theorem 3.2.** *Both DP-OPH-fix and DP-OPH-re in Algorithm 4 achieve* $(\epsilon, \delta)$-*DP.*

*Proof.* Let $\boldsymbol{u}$ and $\boldsymbol{u}'$ be neighbors only differing in one element. Denote $S = \{k \in [K] : h_k(\boldsymbol{u}) \neq h_k(\boldsymbol{u}')\}$ and $S^c = [K] \setminus S$. As discussed before, we can verify that for $k \in S_c$, we have $\frac{P(\tilde{h}_k(\boldsymbol{u})=i)}{P(\tilde{h}_k(\boldsymbol{u}')=i)} = 1$ for any $i = 0, ..., 2^b - 1$. For $k \in S$, $e^{-\epsilon'} \leq \frac{P(\tilde{h}_k(\boldsymbol{u})=i)}{P(\tilde{h}_k(\boldsymbol{u}')=i)} \leq e^{\epsilon'}$ holds for any $i = 0, ..., 2^b - 1$. Thus, for any $Z \in \{0, ..., 2^b - 1\}^K$, the absolute privacy loss can be bounded by

$$\left|\log \frac{P(\tilde{h}(\boldsymbol{u}) = Z)}{P(\tilde{h}(\boldsymbol{u}') = Z)}\right| = \left|\log \prod_{k \in S} \frac{P(\tilde{h}_k(\boldsymbol{u}) = i)}{P(\tilde{h}_k(\boldsymbol{u}') = i)}\right| \leq |S|\epsilon' = |S|\frac{\epsilon}{N}. \quad (6)$$

By Lemma 3.1, with probability $1 - \delta$, $|S| \leq F_{fix}^{-1}(1 - \delta) = N$ for DP-OPH-fix; $|S| \leq F_{re}^{-1}(1 - \delta) = N$ for DP-OPH-re. Hence, (6) is bounded by $\epsilon$ with probability $1 - \delta$. This proves the $(\epsilon, \delta)$-DP. $\quad \square$

**DP-OPH without densification.** From the practical perspective, we may also privatize the OPH without densification (i.e., add DP to the output of Algorithm 2). The first step is to take the last $b$ bits of every non-empty hash and get $K$ hash values from $\{0, ..., 2^b - 1\} \cup \{E\}$. Then, for non-empty bins, we keep the hash value with probability $\frac{e^\epsilon}{e^\epsilon + 2^b - 1}$, and randomly flip it otherwise. For empty bins (i.e., $h_k(\boldsymbol{u}) = E$), we simply assign a random value in $\{0, ..., 2^b - 1\}$ to $\tilde{h}_k(\boldsymbol{u})$. The formal procedure of this so-called DP-OPH-rand method is summarized in Algorithm 5.

**Theorem 3.3.** *Algorithm 5 achieves* $\epsilon$-*DP.*

*Proof.* The proof is similar to the proof of Theorem 3.2. Since the original hash vector $h(\boldsymbol{u})$ is not densified, there only exists exactly one hash value such that $h_k(\boldsymbol{u}) \neq h_k(\boldsymbol{u})$ may happen for

---

**Algorithm 6** Differentially Private Minwise hashing (DP-MH)

---

**Input:** MinHash values $h_1(\boldsymbol{u}), ..., h_K(\boldsymbol{u})$; number of bits $b$; $\epsilon > 0, 0 < \delta < 1$
  $f$: lower bound on the number of non-zeros in each data vector
**Output:** DP-MH values $\tilde{h}(\boldsymbol{u}) = [\tilde{h}_1(\boldsymbol{u}), ..., \tilde{h}_K(\boldsymbol{u})]$

1: Take the last $b$ bits of all hash values $\qquad\qquad\qquad$ ▷ After which $h_k(\boldsymbol{u}) \in \{0, ..., 2^b - 1\}$
2: Set $N = F_{bino}^{-1}(1 - \delta; \frac{1}{f}, K)$, and $\epsilon' = \epsilon/N$
3: **for** $k = 1$ to $K$ **do**
4: $\quad \tilde{h}_k(\boldsymbol{u}) = \begin{cases} h_k(\boldsymbol{u}), & \text{with probability } \frac{e^{\epsilon'}}{e^{\epsilon'}+2^b-1} \\ i, & \text{with probability } \frac{1}{e^{\epsilon'}+2^b-1}, \text{ for } i \in \{0, ..., 2^b - 1\}, \ i \neq h_k(\boldsymbol{u}) \end{cases}$
5: **end for**

---

$\boldsymbol{u}'$ that differs in one element from $\boldsymbol{u}$. W.l.o.g., assume $u_i = 1$ and $u_i' = 0$, and $i \in \mathcal{B}_k$. If bin $k$ is non-empty for both $\boldsymbol{u}$ and $\boldsymbol{u}'$ (after permutation), then for any $Z \in \{0, ..., 2^b - 1\}^K$, $\left| \log \frac{P(\tilde{h}(\boldsymbol{u})=Z)}{P(\tilde{h}(\boldsymbol{u}')=Z)} \right| \leq \epsilon$ according to our analysis in Theorem 3.2 (the probability of hash in $[K] \setminus \{k\}$ cancels out). If bin $k$ is empty for $\boldsymbol{u}'$, since $1 \leq \frac{e^\epsilon}{e^\epsilon+2^b-1} / \frac{1}{2^b} \leq e^\epsilon$ and $e^{-\epsilon} \leq \frac{1}{2^b} / \frac{1}{e^\epsilon+2^b-1} \leq 1$, we also have $\left| \log \frac{P(\tilde{h}(\boldsymbol{u})=Z)}{P(\tilde{h}(\boldsymbol{u}')=Z)} \right| \leq \epsilon$. Therefore, the algorithm is $\epsilon$-DP. $\qquad\square$

Compared with Algorithm 4, DP-OPH-rand achieves strict DP with smaller flipping probability (effectively, $N \equiv 1$ in Algorithm 4). This demonstrates the essential benefit of "binning" in OPH, since the change in one data coordinate will only affect one hash value (if densification is not applied). As a consequence, the non-empty hash values are less perturbed in DP-OPH-rand than in DP-OPH-fix or DP-OPH-re. However, this comes with an extra cost as we have to assign random bits to empty bins, which do not provide any useful information about the data. Moreover, this extra cost does not diminish as $\epsilon$ increases, because the number of empty bins only depends on the data itself and $K$.

### 3.3 COMPARISON WITH DIFFERENTIALLY PRIVATE MINHASH (DP-MH)

While we have presented our main contributions on the DP-OPH algorithms, we also discuss the DP MinHash (DP-MH) method (Algorithm 6) as a baseline comparison. The general mechanism of DP-MH is the same as densified DP-OPH . The main difference between Algorithm 6 and Algorithm 4 is in the calculation of $N$. In Algorithm 6, we set $N = F_{bino}^{-1}(1 - \delta; \frac{1}{f}, K)$ where $F_{bino}^{-1}(x; p, n)$ is the inverse cumulative mass function of $Binomial(p, n)$ with $n$ trials and success probability $p$.

**Theorem 3.4.** *Algorithm 6 is $(\epsilon, \delta)$-DP.*

*Proof.* We use the same proof strategy for Theorem 3.2 by noting that $X = \sum_{k=1}^{K} \mathbb{1}\{h_k(\boldsymbol{u}) \neq h_k(\boldsymbol{u}')\}$ for neighboring $\boldsymbol{u}$ and $\boldsymbol{u}'$ follows $Binomial(\frac{1}{f}, K)$. $\qquad\square$

In a related work, Aumüller et al. (2020) also proposed to apply randomized response to MinHash. However, the authors incorrectly used a tail bound for the binomial distribution (see their Lemma 1) which is only valid for small deviation. In DP, $\delta$ is often very small (e.g., $10^{-6}$), so the large deviation tail bound should be used which is looser than the one used therein[1]. That said, in their paper, the perturbation is underestimated and their method does not satisfy DP rigorously. In our Algorithm 6, we fix this minor error by using the exact probability mass function to compute the tail probability, which also avoids any loss due to the concentration bounds.

**Comparison: Densified DP-OPH versus DP-MH.** We compare $N$, the "privacy discount factor", in DP-OPH-fix, DP-OPH-re, and DP-MH. Smaller $N$ leads to smaller bit flipping probability which benefits the utility. In Figure 1, we plot $N$ vs. $f$, for $D = 1024$, $K = 64$, and $\delta = 10^{-6}$. Similar comparison also holds for other $D, K$ combinations. From the figure, we observe that $N$ in DP-OPH is typically smaller than that in DP-MH, showing the advantages of OPH from the privacy perspective. Moreover, $N$ for DP-OPH-re is consistently smaller than that for DP-OPH-fix. This

---

[1]For $X$ following a Binomial distribution with mean $\mu$, Aumüller et al. (2020) used the concentration inequality $P(X \geq (1 + \xi)\mu) \leq \exp(-\frac{\xi^2 \mu}{3})$, which only holds when $0 \leq \xi \leq 1$. For large deviations (large $\xi$), the valid Binomial tail bound should be $P(X \geq (1 + \xi)\mu) \leq \exp(-\frac{\xi^2 \mu}{\xi+2})$.

illustrates that re-randomization in the densification process is an important step to ensure better privacy. A comparison of the MSE of the unbiased Jaccard estimators are placed in Appendix A.

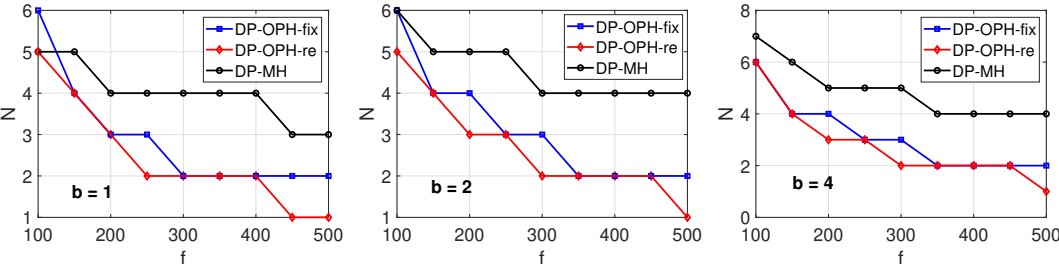

Figure 1: Comparison of the privacy discount factor $N$ for densified DP-OPH and DP-MH, against the number of non-zero elements in the data vector $f$. $D = 1024, K = 64, \delta = 10^{-6}$.

## 4 EXPERIMENTS

We conduct retrieval experiments on three public datasets from various domains: (1) Leukemia gene expression dataset (https://sbcb.inf.ufrgs.br/cumida); (2) MNIST (LeCun et al., 1998) hand-written digit dataset; (3) Webspam (Chang & Lin, 2011) dataset for spam detection. All the datasets are binarized to 0/1. For Leukemia, we first standardize the features columns (to mean 0 and std 1), and then keep entries larger than 1 to be 1 and zero out the others. For MNIST and Webspam, we simply set the non-zero entries to 1. For Leukemia, since the data size is small, we treat every data point as a query and other points as the database. For MNIST and Webspam, we use the train set as the database, and the test set as queries. For each query point, we set the ground truth ("gold-standard") neighbors as the top 50 data points in the database with highest Jaccard similarity to the query. To search with DP-OPH and DP-MH, we generate the private hash values and compute the collision estimator (2) between the query and each data point. Then, we retrieve the data points with the highest estimated Jaccard similarity to the query. For densified DP-OPH (Algorithm 4) and DP-MH (Algorithm 6), we ensure the lower bound $f$ on the number of non-zero elements by filtering the data points with at least $f$ non-zeros. We use $f = 1000, 50, 500$ for Leukemia, MNIST, and Webspam, respectively, which cover $100\%, 99.9\%, 90\%$ of the total data points. We average the precision and recall over all the query points and over 5 independent runs.

**Results.** In Figure 2, we report the results for Leukemia with $b = 1, 2, 4$ and $\epsilon \in [1, 50]$. Due to space limitation, we plot the precision here; the recall comparisons are similar. We see that:

- DP-OPH-re performs considerably better than DP-MH and DP-OPH-fix, for all $\epsilon$ levels.

- DP-OPH-rand achieves good accuracy with small $\epsilon$ (e.g., $\epsilon < 5$), but stops improving with $\epsilon$ afterwards (due to the random bits for the empty bins), which demonstrates the trade-off discussed in Section 3.2. When $\epsilon$ gets larger (e.g., $\epsilon > 10$), DP-OPH-re performs the best.

- Increasing $b$ is relatively more beneficial for DP-OPH-rand as it can achieve higher search accuracy with small $\epsilon$. Also, larger $\epsilon$ is required for DP-OPH-re to bypass DP-OPH-rand.

The results on MNIST and Webspam are presented in Figure 3 and Figure 4, respectively. Similarly, DP-OPH-re achieves better performance than DP-MH and DP-OPH-fix for all $\epsilon$. DP-OPH-rand performs the best with $\epsilon < 10$. Yet, it is outperformed by DP-OPH-re with larger $\epsilon$.

## 5 CONCLUSION

In this paper, we propose differential private one permutation hashing (DP-OPH). We develop three variants depending on the densification procedure of OPH, and provide detailed derivation and privacy analyses of our algorithms. We show the advantage of the proposed DP-OPH over the DP Min-Hash alternative for hashing the Jaccard similarity. Experiments are conducted on retrieval tasks to justify the effectiveness of the proposed DP-OPH, and provide guidance on the appropriate choice of the DP-OPH variant in different scenarios. In Appendix B, we also provide DP-BCWS which is based on bin-wise consistent weighted samples (BCWS) (Li et al., 2019) for weighted Jaccard similarity (for non-negative data). Given the efficiency and strong performance, we expect DP-OPH to be a useful private hashing method in practical applications.

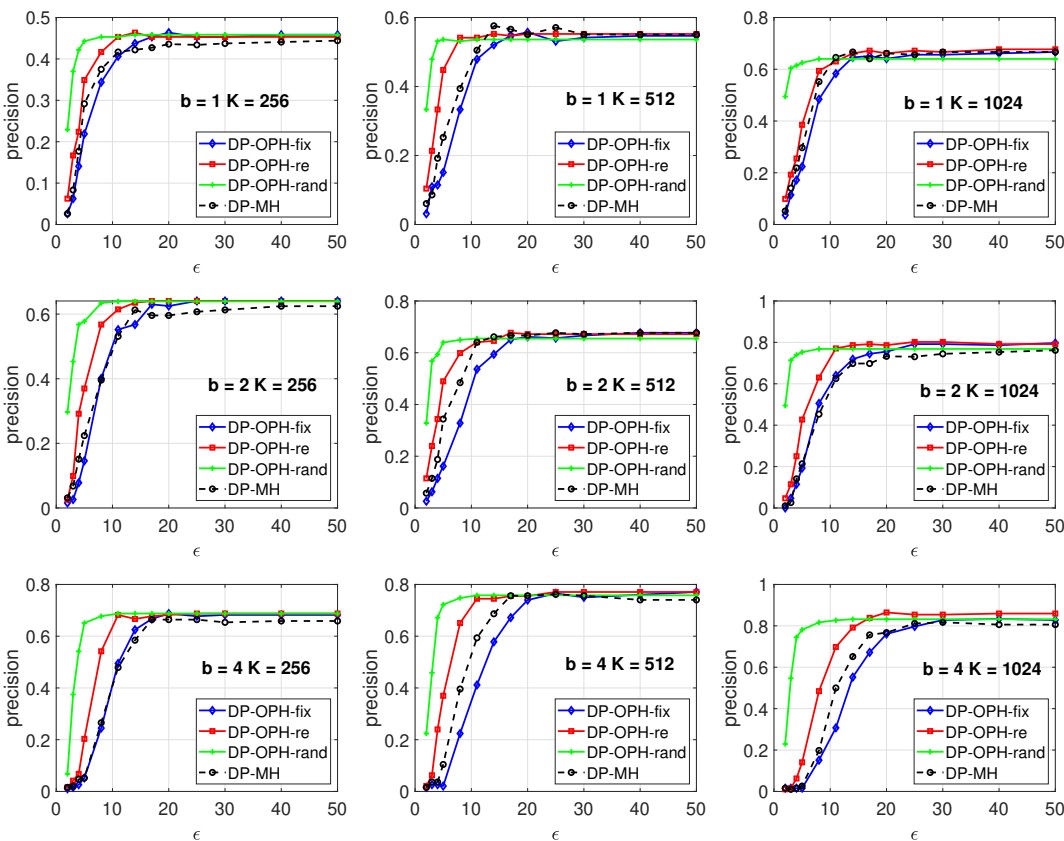

Figure 2: Precision@1 results on Leukemia gene expression dataset with $b = 1, 2, 4$. $\delta = 10^{-6}$. We check the precision to recover the most similar neighbor for every data point.

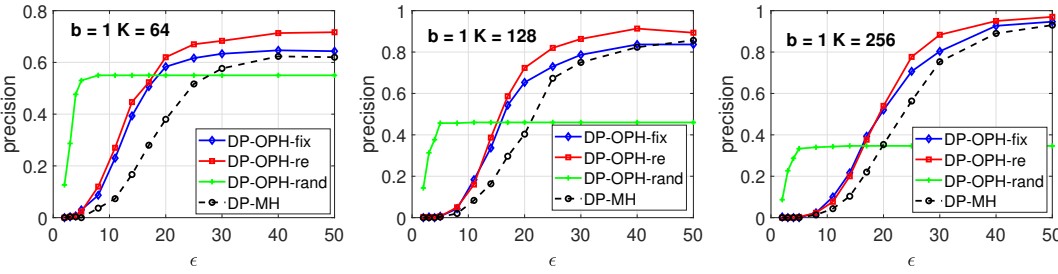

Figure 3: Precision@10 results on MNIST dataset for $b = 1$. $\delta = 10^{-6}$.

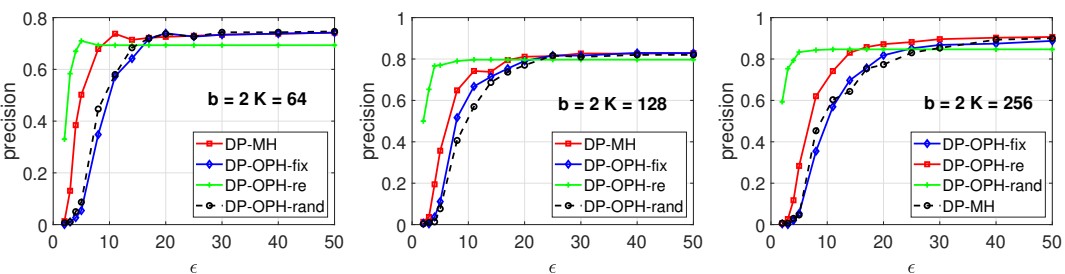

Figure 4: Precision@10 results on Webspam dataset with $b = 2$. $\delta = 10^{-6}$.

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

## A    UNBIASED JACCARD ESTIMATOR AND THE MSE COMPARISON

In the main paper, we have shown that DP-OPH-re has the smallest "privacy discount factor" $N$ compared to DP-OPH-fix and DP-MH (Figure 1). Here we further compare there Jaccard estimation accuracy. For the two densified DP-OPH variants, DP-OPH-fix and DP-OPH-re, and the DP MinHash (DP-MH) methods, each full-precision (and unprivatized) hash value of $h(\boldsymbol{u})$ and $h(\boldsymbol{v})$ has collision probability equal to $P(h(\boldsymbol{u}) = h(\boldsymbol{v})) = J(\boldsymbol{u}, \boldsymbol{v})$. Let $h^{(b)}(\boldsymbol{u})$ denote the $b$-bit hash values. Since we assume the last $b$ bits are uniformly assigned, we have $P(h^{(b)}(\boldsymbol{u}) = h^{(b)}(\boldsymbol{v})) = J + (1 - J)\frac{1}{B}$. Denote $p = \frac{\exp(\epsilon/N)}{\exp(\epsilon/N) + 2^b - 1}$. By simple probability calculation, the privatized $b$-bit hash values has collision probability

$$P(\tilde{h}(\boldsymbol{u}) = \tilde{h}(\boldsymbol{v}))$$
$$= P(\tilde{h}(\boldsymbol{u}) = \tilde{h}(\boldsymbol{v})|h^{(b)}(\boldsymbol{u}) = h^{(b)}(\boldsymbol{v}))P(h^{(b)}(\boldsymbol{u}) = h^{(b)}(\boldsymbol{v}))$$
$$+ P(\tilde{h}(\boldsymbol{u}) = \tilde{h}(\boldsymbol{v})|h^{(b)}(\boldsymbol{u}) \neq h^{(b)}(\boldsymbol{v}))P(h^{(b)}(\boldsymbol{u}) \neq h^{(b)}(\boldsymbol{v}))$$
$$= \left[p^2 + \frac{(1-p)^2}{2^b - 1}\right]\left(\frac{1}{2^b} + \frac{2^b - 1}{2^b}J\right) + \left[\frac{2p(1-p)}{2^b - 1} + \frac{2^b - 2}{(2^b - 1)^2}(1-p)^2\right]\left(\frac{2^b - 1}{2^b} - \frac{2^b - 1}{2^b}J\right),$$

which implies $J = \frac{(2^b - 1)(2^b P(\tilde{h}(u) = \tilde{h}(v)) - 1)}{(2^b p - 1)^2}$. Therefore, let $\hat{J} = \frac{1}{K}\sum_{k=1}^{K} \mathbb{1}\{\tilde{h}_k(u) = \tilde{h}_k(v)\}$, then an unbiased estimator of $J$ is

$$\hat{J}_{unbias} = \frac{(2^b - 1)(2^b \hat{J} - 1)}{(2^b p - 1)^2}. \tag{7}$$

To compare the mean squared error (MSE), we simulate a two data vectors with $D = 1024$, $K = 64$, and $J = 1/3$. In Figure 5, we vary $f$, the number of non-zeros per data vector, and report the empirical MSE of the unbiased estimator (7) for DP-OPH-fix, DP-OPH-re and DP-MH, respectively. As we can see, the comparison is consistent with the comparison of $N$ in Figure 1, that the proposed DP-OPH-re has smallest MSE among the three competitors. This again justifies the advantage of DP-OPH-re with re-randomized densification.

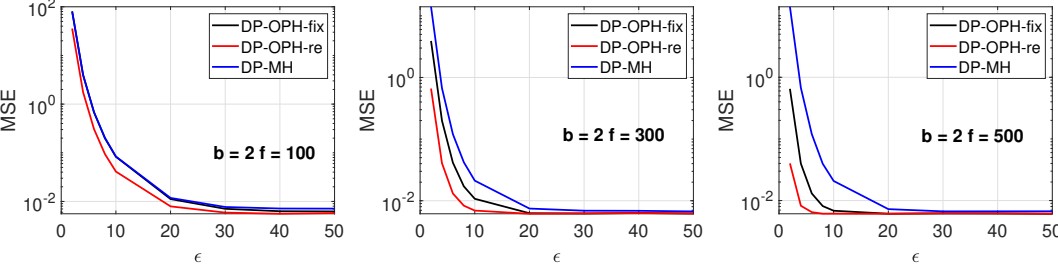

Figure 5: Empirical MSE comparison of the unbiased Jaccard estimator (7) from DP-OPH-fix, DP-OPH-re and DP-MH. $D = 1024, K = 64, \delta = 10^{-6}$.

## B    EXTENSION: DIFFERENTIALLY PRIVATE BIN-WISE CONSISTENT WEIGHTED SAMPLING (DP-BCWS) FOR WEIGHTED JACCARD SIMILARITY

In our main paper, we focused on DP hashing algorithms for the binary Jaccard similarity. Indeed, our algorithm can also be extended to hashing the weighted Jaccard similarity: (recall the definition)

$$J_w(\boldsymbol{u}, \boldsymbol{v}) = \frac{\sum_{i=1}^{D} \min\{u_i, v_i\}}{\sum_{i=1}^{D} \max\{u_i, v_i\}}, \tag{8}$$

for two non-negative data vectors $\boldsymbol{u}, \boldsymbol{v} \in \mathbb{R}_+$. The standard hashing algorithm for (8) is called Consistent Weighted Sampling (CWS) as summarized in Algorithm 7 (Ioffe, 2010; Manasse et al.,

---

**Algorithm 7** Consistent Weighted Sampling (CWS)

---

**Input:** Non-negative data vector $\boldsymbol{u} \in \mathbb{R}_+^D$
**Output:** Consistent weighted sampling hash $h^* = (i^*, t^*)$

1: **for** every non-zero $v_i$ **do**
2:      $r_i \sim Gamma(2,1), \ c_i \sim Gamma(2,1), \ \beta_i \sim Uniform(0,1)$
3:      $t_i \leftarrow \lfloor \frac{\log u_i}{r_i} + \beta_i \rfloor, \ y_i \leftarrow \exp(r_i(t_i - \beta_i))$
4:      $a_i \leftarrow c_i / (y_i \exp(r_i))$
5: **end for**
6: $i^* \leftarrow arg\min_i \ a_i, \qquad t^* \leftarrow t_{i^*}$

---

2010; Li et al., 2021). To generate one hash value, we need three length-$D$ random vectors $\boldsymbol{r} \sim Gamma(2,1)$, $\boldsymbol{c} \sim Gamma(2,1)$ and $\boldsymbol{\beta} \sim Uniform(0,1)$. We denote Algorithm 7 as a function $CWS(\boldsymbol{u}; \boldsymbol{r}, \boldsymbol{c}, \boldsymbol{\beta})$. Li et al. (2019) proposed bin-wise CWS (BCWS) which exploits the same idea of binning as in OPH. The binning and densification procedure of BCWS is exactly the same as OPH (Algorithm 2 and Algorithm 3), except that every time we apply CWS, instead of MinHash, to the data in the bins to generate hash values. Note that in CWS, the output contains two values: $i^*$ is a location index similar to the output of OPH, and $t^*$ is a real-value scalar. Prior studies (e.g., Li et al. (2021)) showed that the second element has minimal impact on the estimation accuracy in most practical cases (i.e., only counting the collision of the first element suffices). Therefore, in our study, we only keep the first integer element as the hash output for subsequent learning tasks.

For weighted data vectors, we follow the prior DP literature on weighted sets (e.g., Xu et al. (2013); Smith et al. (2020); Dickens et al. (2022); Zhao et al. (2022)) and define the neighboring data vectors as those who differ in one element. To privatize BCWS, there are also three possible ways depending on the densification option. Since the DP algorithm design for densified BCWS requires rigorous and non-trivial computations which might be an independent study, here we empirically test the ($b$-bit) DP-BCWS method with random bits for empty bins. The details are provided in Algorithm 8. In general, we first randomly split the data entries into $K$ equal length bins, and apply CWS to the data $\boldsymbol{u}_{\mathcal{B}_k}$ in each non-empty bin $\mathcal{B}_k$ using the random numbers $(\boldsymbol{r}_{\mathcal{B}_k}, \boldsymbol{c}_{\mathcal{B}_k}, \boldsymbol{\beta}_{\mathcal{B}_k})$ to generated $K$ hash values (possibly including empty bins). After each hash is truncated to $b$ bits, we uniformly randomly assign a hash value in $\{0, ..., 2^b - 1\}$ to every empty bin.

---

**Algorithm 8** Differential Private Bin-wise Consistent Weighted Sampling (DP-BCWS)

---

**Input:** Binary vector $\boldsymbol{u} \in \{0,1\}^D$; number of hash values $K$; number of bits per hash $b$
**Output:** DP-BCWS hash values $\tilde{h}_1(\boldsymbol{u}), ..., \tilde{h}_K(\boldsymbol{u})$

1: Generate length-$D$ random vectors $\boldsymbol{r} \sim Gamma(2,1), \ \boldsymbol{c} \sim Gamma(2,1), \ \boldsymbol{\beta} \sim Uniform(0,1)$
2: Let $d = D/K$. Use a permutation $\pi : [D] \mapsto [D]$ with fixed seed to randomly split $[D]$ into $K$ equal-size bins $\mathcal{B}_1, ..., \mathcal{B}_K$, with $\mathcal{B}_k = \{j \in [D] : (k-1)d + 1 \leq \pi(j) \leq kd\}$
3: **for** $k = 1$ to $K$ **do**
4:      **if** Bin $\mathcal{B}_k$ is non-empty **then**
5:          $h_k(\boldsymbol{u}) \leftarrow CWS(\boldsymbol{u}_{\mathcal{B}_k}; \boldsymbol{r}_{\mathcal{B}_k}, \boldsymbol{c}_{\mathcal{B}_k}, \boldsymbol{\beta}_{\mathcal{B}_k})$         ▷ Run CWS within each non-empty bin
6:          $h_k(\boldsymbol{u}) \leftarrow$ last $b$ bits of $h_k(\boldsymbol{u})$
7:          $\tilde{h}_k(\boldsymbol{u}) = \begin{cases} h_k(\boldsymbol{u}), & \text{with probability } \frac{e^\epsilon}{e^\epsilon + 2^b - 1} \\ i, & \text{with probability } \frac{1}{e^{\epsilon'} + 2^b - 1}, \text{ for } i \in \{0, ..., 2^b - 1\}, \ i \neq h_k(\boldsymbol{u}) \end{cases}$
8:      **else**
9:          $h_k(\boldsymbol{u}) \leftarrow E$
10:         $\tilde{h}_k(\boldsymbol{u}) = i$ with probability $\frac{1}{2^b}$, for $i = 0, ..., 2^b - 1$  ▷ Assign random bits to empty bin
11:      **end if**
12: **end for**

---

Using the same proof arguments as Theorem 3.3, we have the following guarantee.

**Theorem B.1.** *Algorithm 8 satisfies $\epsilon$-DP.*

**Empirical evaluation.** In Figure 6, we train an $l_2$-regularized logistic regression on the DailySports dataset[2]. and report the test accuracy with various $b$ and $K$ values. The $l_2$ regularization parameter $\lambda$ is tuned over a fine grid from $10^{-4}$ to 10. Similar to the results in the previous section, the performance of DP-BCWS becomes stable as long as $\epsilon > 5$. Note that, linear logistic regression only gives $\approx 75\%$ accuracy on original DailySports dataset (without DP). With DP-BCWS, the accuracy can reach $\approx 98\%$ with $K = 1024$ and $\epsilon = 5$.

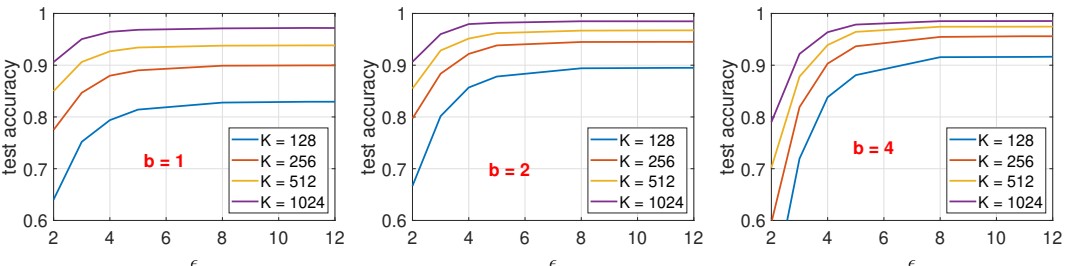

Figure 6: Test classification accuracy of DP-BCWS on DailySports dataset (Asuncion & Newman, 2007) with $l_2$-regularized logistic regression.

In Figure 7, we train a neural network with two hidden layers of size 256 and 128 respectively on MNIST. We use the ReLU activation function and the standard cross-entropy loss. We see that, in a reasonable privacy regime (e.g., $\epsilon < 10$), DP-BCWS is able to achieve $\approx 95\%$ test accuracy with proper $K$ and $b$ combinations (one can choose the values depending on practical scenarios and needs). For example, with $b = 4$ and $K = 128$, DP-BCWS achieves $\approx 97\%$ accuracy at $\epsilon = 8$.

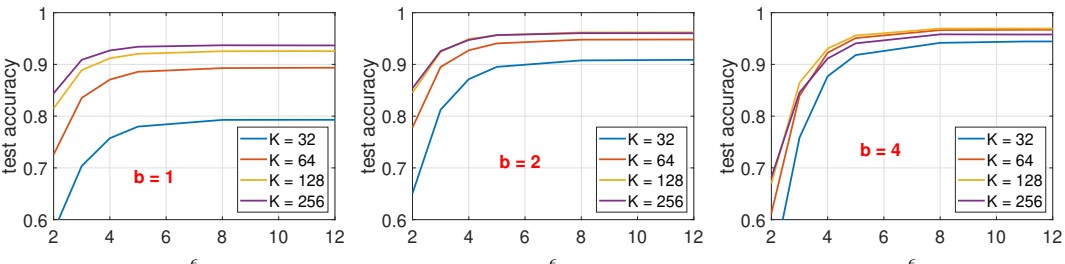

Figure 7: Test classification accuracy of DP-BCWS on MNIST with 2-hidden layer neural network.

## C  PROOF OF LEMMA 3.1

**Lemma C.1** (Li et al. (2012)). *Let $f = |\{i : u_i = 1\}|$, and $I_{emp,k}$ be the indicator function that the $k$-th bin is empty, and $N_{emp} = \sum_{k=1}^{K} I_{emp,k}$. Suppose $mod(D, K) = 0$. We have*

$$P\left(N_{emp} = j\right) = \sum_{\ell=0}^{K-j} (-1)^\ell \binom{K}{j} \binom{K-j}{\ell} \binom{D(1-(j+\ell)/K)}{f} \Big/ \binom{D}{f}.$$

**Lemma C.2** (Li et al. (2019)). *Conditional on the event that $m$ bins are non-empty, let $\tilde{f}$ be the number of non-zero elements in a non-empty bin. Denote $d = D/K$. The conditional probability distribution of $\tilde{f}$ is given by*

$$P\left(\tilde{f} = j \big| m\right) = \frac{\binom{d}{j} H(m-1, f-j|d)}{H(m, f|d)}, \quad j = \max\{1, f-(m-1)d\}, ..., \min\{d, f-m+1\},$$

---

[2] https://archive.ics.uci.edu/ml/datasets/daily+and+sports+activities

*where $H(\cdot)$ follows the recursion: for any $0 < k \le K$ and $0 \le n \le f$,*

$$H(k,n|d) = \sum_{i=\max\{1,n-(k-1)d\}}^{\min\{d,n-k+1\}} \binom{d}{i} H(k-1,n-i|d), \quad H(1,n|d) = \binom{d}{n}.$$

*Proof.* (of Lemma 3.1) Without loss of generality, suppose $\boldsymbol{u}$ and $\boldsymbol{u}'$ differ in the $i$-th dimension, and by the symmetry of DP, we can assume that $u_i = 1$ and $u_i' = 0$. We know that $i$ is assigned to the $\lceil mod(\pi(i),d) \rceil$-th bin. Among the $K$ hash values, this change will affect all the bins that uses the data/hash of the $k^* = \lceil mod(\pi(i),d) \rceil$-th bin (after permutation), both in the first scan (if it is non-empty) and in the densification process. Let $N_{emp}$ be the number of empty bins in $h(\boldsymbol{u})$, and $\tilde{f}$ be the number of non-zero elements in the $k^*$-th bin. We have, for $x = 0, ..., K - \lceil f/d \rceil$,

$$P(X = x) = \sum_{j=\max(0,K-f)}^{K-\lceil f/d \rceil} \sum_{z=1}^{\min(f,d)} P\left(X = x \Big| \tilde{f} = z, N_{emp} = j\right) P\left(\tilde{f} = z, N_{emp} = j\right)$$

$$= \sum_{j=\max(0,K-f)}^{K-\lceil f/d \rceil} \sum_{z=1}^{\min(f,d)} P\left(X = x \Big| \tilde{f} = z, N_{emp} = j\right) P\left(\tilde{f} = z | K - j\right) P\left(N_{emp} = j\right),$$

where $P\left(\tilde{f} = z | K - j\right)$ is given in Lemma C.2 and $P\left(N_{emp} = j\right)$ can be calculated by Lemma C.1. To compute the first conditional probability, we need to compute the number of times the $k^*$-th bin is picked to generated hash values, and the hash values are different for $\boldsymbol{u}$ and $\boldsymbol{u}'$. Conditional on $\{\tilde{f} = z, N_{emp} = j\}$, denote $\Omega = \{k : \mathcal{B}_k \text{ is empty}\}$, and let $R_k$ be the non-empty bin used for the $k$-th hash value $h_k(\boldsymbol{u})$, which takes value in $[K] \setminus \Omega$. We know that $|\Omega| = j$. We can write

$$X = \mathbb{1}\{h_{k^*}(\boldsymbol{u}) \neq h_{k^*}(\boldsymbol{u}')\} + \sum_{k \in \Omega} \mathbb{1}\{R_k = k^*, h_k(\boldsymbol{u}) \neq h_k(\boldsymbol{u}')\}.$$

Here we separate out the first term because the $k^*$-th hash always uses the $k^*$-bin. Note that the densification bin selection is uniform, and the bin selection is independent of the permutation for hashing. For the fixed densification, since the hash value $h_{k^*}(\boldsymbol{u})$ is generated and used for all hash values that use $\mathcal{B}_{k^*}$, we have

$$P\left(X = x \Big| \tilde{f} = z, N_{emp} j\right) = \mathbb{1}\{x = 0\}\left(1 - P_{\neq}\right) + \mathbb{1}\{x > 0\}P_{\neq} \cdot g_{bino}\left(x - 1; \frac{1}{K - j}, j\right),$$

where $g_{bino}(x; p, n)$ is the probability mass function of the binomial distribution with $n$ trials and success rate $p$, and $P_{\neq} = P(h_{k^*}(\boldsymbol{u}) \neq h_{k^*}(\boldsymbol{u}')) = \left(1 - \frac{1}{2^b}\right)\frac{1}{z}$. Based on the same reasoning, for re-randomized densification, we have

$$P\left(X = x \Big| \tilde{f} = z, N_{emp} j\right) = (1 - P_{\neq}) \cdot g_{bino}\left(x; \frac{P_{\neq}}{K - j}, j\right) + P_{\neq} \cdot g_{bino}\left(x - 1; \frac{P_{\neq}}{K - j}, j\right).$$

Combining all the parts together completes the proof. $\qquad\square$

