# OpenReview forum: "Differentially Private One Permutation Hashing"
_ICLR.cc/2024/Conference — Submitted to ICLR 2024_

### Official Review · Reviewer_zun1 · 2023-10-27

**Soundness:** 3 good
**Presentation:** 2 fair
**Contribution:** 2 fair
**Rating:** 5
**Confidence:** 3

**Summary:**

This paper studies differentially private minwise hashing (MinHash) algorithms as variants of the one permutation hashing (OPH) algorithm. Three DP variants are introduced in the paper, DP-OPH-fix, DP-OPH-re and DP-OPH-rand. The authors show that these variants can provide attribute-level DP guarantee with in-depth examination and utilization of the nature of OPH. Some experiments show that the proposed three variants can outperform the straw-man algorithm (DP-MH), and DP-OPH-rand and DP-OPH-re have advantages in different privacy strengths.

**Strengths:**

1. The paper designs three different variants of differentially private minwise hashing (MinHash) algorithms.
2. The proof for deciding how the privacy budget should be split demonstrates an in-depth analysis of the nature of OPH, and it can be inspiring for the study of DP on other sketches.
3. The paper provides theoretical analysis and some empirical evidence to support the superiority of the proposed algorithm.

**Weaknesses:**

1. The attributed-level DP seems to relatively weak in terms of privacy protection. For example, when one attribute in MNIST data used in the experiment is just whether a pixel value is zero, ensuring such attribute level indistinguishability may provide only limited privacy protection.
2. While there is no theoretical utility guarantee shown, it is also hard to evaluate the empirical effectiveness of the proposed algorithm because (a) the non-private baseline is missing (b) it is not clear what it means to downstream machine learning tasks given such precision.
3. While attribute-level DP is already a weaker privacy notion, the empirical results show that it requires $\epsilon>5$ to achieve some acceptable precision. Also, any $\epsilon > 20$ can provide extremely limited privacy protection and usually will not be considered in privacy literature.
4. The writing of the paper can be improved. For example, $\tilde{f}$ in equation (3) and $N_{emp}$ are never introduced in the main text, which harm the integrity of  the main text.

**Questions:**

1. What is the non-private OPH performance on the same datasets?
2. How useful are the retrieved samples to the downstream tasks with the shown precision?
3. Is it possible to strengthen the protocol to be sample-level DP? What may be the key challenge?
4. Is there any theoretical performance guarantee for the DP OPH algorithms?
5. How tight is the proposed privacy splitting compared to the existing sequential composition techniques (e.g., Renyi DP)?

---

> ### Author Response · Authors · 2023-11-18
> **Thank you**
>
> Dear Reviewer zun1:
>
> Thanks for your review of our paper. Since the points in "Weakness" (except W3) overlap with the "Questions" to a large extent, we will combine some of our replies to "Weakness" and "Questions". Thank you.
>
> W3. On the value of $\epsilon$: DP has been deployed in a wide range of fields and applications, and the feasible $\epsilon$ depends on the specific use case. For example, in the tutorial [How to dp-fy ml: A
> practical guide to machine learning with differential privacy, Ponomareva et al., 2023, Section 5.2] by Google, the researchers defined $\epsilon<10$ as "reasonable privacy protection". For the U.S Census use case [The use of differential privacy for census data and its
> impact on redistricting: The case of the 2020 U.S. Census, Kenny et al., 2021], $\epsilon=12.2$ is chosen. The PyTorch tutorial of DP-SGD training set $\epsilon=47.2$ (https://github.com/pytorch/opacus/blob/main/tutorials/building_image_classifier.ipynb). Therefore, depending on the application and privacy requirement, we believe that different $\epsilon$ values in DP-OPH are useful in difference use cases.
>
>
> Q1 & W2. On the non-private performance: The precision at $\epsilon=50$ (after the curves become flat) essentially equals to the precision of the non-private methods.
>
>
> Q2 & W2. On the implication of experiments: the task in our experiments is nearest neighbor search with hashing. This is a standard task to evaluate the empirical performance of hashing methods, which demonstrates the utility of different algorithms in terms of Jaccard similarity approximation. Also, nearest neighbor search is itself an important "downstream task" of the hash samples in industrial applications where hashing/sketching methods are widely used for accelerated computation. Thus, our results already show that the proposed methods is advantageous in hashing applications with privacy constraints. We presented our results on three datasets from different domains: biology, image, and web. We hope this is sufficient to justify the benefits of our proposed methods for a variety of applications.
>
>
> Q3 & W1. On the privacy definition: As we mentioned/cited in the paper, attribute-level DP is a standard setup in the literature of private sketching/hashing, e.g., [Kenthapadi et al, 2013], [Stausholm, 2021], [Zhao et al, 2022], [Smith et al, 2020], [Dickens et al, 2022]. The reason is that, our goal is to privatize the output hash values, and they are representations (or called "signatures" in some fields) of each data vector. Thus, it is natural to treat a data vector as the "database", and define "neighboring" on the coordinate level.
>
> We have two remarks here regarding Question 3. First, for real data vectors and when the randomness of the hashing algorithm is internal and private [Blocki et al.2012], there is a privacy definition that requires $\|u-u'\|\leq 1$ for neighboring vectors $u$ and $u'$. For binary data, this is equivalent to our definition. We may extend the definition to "differing in multiple coordinates", which is similar to the notion of "group privacy". This requires more involved analysis, which could be an independent study for future investigation. In this paper, we follow the common definition of "one coordinate change" in most related works on private sketching.
>
> Second, if what the reviewer suggested is to protect arbitrarily different vectors $u$ and $u'$, then the problem becomes impossible to solve because the output hash values of two arbitrary vectors in $u$ and $u'$ in $R^d$ can also be arbitrarily different. It is hard to find useful data representation while requiring that the representation of any data vector should look similar (which is required by DP).

---

> > ### Author Response · Authors · 2023-11-18
> > **Author Rebuttal (Part 2)**
> >
> > Q4 and W2. On the theoretical utility: As you kindly pointed out, in principle the variance of the (unbiased) Jaccard estimator from DP-OPH can be computed. However, given the already very complicated combinatorial form of the variance of non-private OPH, the expression will be extremely sophisticated also considering the b-bit coding and DP flipping. Thus, a straightforward analytical comparison is hard. In Appendix A of our submission, we present the empirical MSE comparisons of different DP hashing approaches, and we hope these are convincing enough to show the advantages of DP-OPH. Moreover, we know that in general the extra variation brought by DP flipping would be largely determined by $N$, the privacy discount factor in DP-OPH-re and DP-OPH-fix. In Figure 1, we compare the $N$ of difference methods to intuitively show the advantage of the proposed approach in terms of privacy. These comparisons, together with the extensive experiments, show that the proposed algorithms provide better utility.
> >
> > W4. $\tilde f$ is the number of non-zeros in a non-empty bin, which is defined in a lemma in the Appendix. We will clarify in the main paper. Thank you.
> >
> > Q5. On alternative composition theorems: The analysis of our work is based on two parts: (1) establish a high probability event (with probability $1-\delta$) that the number of changed hashes is smaller than $N$ between any two neighboring data; (2) apply DP composition to the individually pure $\epsilon'$-DP mechanisms for the $N$ hashes. It is known that if in step (2) we don't want to invoke additional failure probability $\delta'$ and stick with pure $\epsilon$-DP after composition, then the basic composition applied in our method is optimal, see, for example, a nice summary and tutorial in [1].
> >
> > [1] Composition of Differential Privacy and Privacy Amplification by Subsampling, Steinke 2022.
> >
> > Alternatively, we may use other composition theorems, like the advanced composition, or composition theorems induced by concentrated DP or Renyi DP as you kindly mentioned. We have two remarks on their comparisons here: (i) These advanced composition theorems may not be better than the basic composition, especially when the number of composition elements is not very large, e.g., Figure 1 in [1] (for DP-OPH, this corresponds to the case where the data is not extremely sparse, which is common in practice); (2) In these methods, composing $\epsilon'$-DP mechanisms would introduce an extra $\delta'$ in $(\epsilon,\delta)$-DP which needs to be counted in the total budget of $\delta$ in addition to the $\delta$ derived in step (1). Therefore, for simplicity and convenience, in our methods we adopted the basic composition.
> >
> > We take your suggestion and agree that a brief discussion on this would be helpful. We will include a remark on these alternative choices for DP composition in the paper.
> >
> > Thanks again for your efforts in reviewing our work. We hope our response well addresses your questions. Please kindly let us know if more clarification is needed. Thank you.

---

> > > ### Comment · Reviewer_zun1 · 2023-11-22
> > >
> > > Thanks for the response from the authors. However, the privacy-utility trade-off of this paper still seems weaker than that of well-known existing DP works. Personally speaking, comparing with the PyTorch tutorial example is inappropriate, as it was designed to demo how their DP package works and is easy for users to debug. More important, even the DP-SGD demo example has a stronger privacy definition than the one in this paper, as they use **per-sample** DP while this paper employs **per-pixel** DP, which means if naively apply "group privacy", the demo can provide 47.2/(3x32x32) per-pixel DP.

---

> > > > ### Author Response · Authors · 2023-11-23
> > > > **Reply to the additional comments**
> > > >
> > > > Dear Reviewer zun1,
> > > >
> > > > Thanks for the reply. Our proposed methods in general approach the perormance of non-private counterparts when $\epsilon=5\sim 15$.
> > > >
> > > > In the PyTorch official tutorial on DP-SGD, we see that even with $\epsilon=47$, the resnet accuracy is still way lower than the non-private model. While this is not a formal publication, we think it might still be a good example that $\epsilon$ should really depend on different privacy settings and use cases.
> > > >
> > > > Also, as we mentioned in the rebuttal, the problem of our work (or more generally, many DP sketching methods) is different from many other "algorithm-specific" DP methods, because hashing/sketching is a data-level (sample-wise) privacy mechanism. That said, we find private representations for **every** data sample. Those representations can be used for large-scale search and learning also with DP guarantees. This is in general a harder problem than reporting the summary statistics, for example, the mean and median of a dataset, or the aggregated gradients over a mini-batch as in DP-SGD. If we want to find a conceptually analogous setting in the language of DP-SGD, our privacy model should correspond to DP-SGD with mini-batch size 1 (we protect and use the gradient of every single data sample). In that case, the aggregated DP noise loses the $1/B$ factor (where $B$ is the batch size) and the problem becomes more difficult---we will need an $\epsilon$ even much larger than $47$ to achieve acceptable utility in that pytorch example. We hope our reply helps the reviewer better understand the privacy setting and application scenarios of our work.
> > > >
> > > > We again sincerely appreciate the reviewer for the review and discussion. Thank you and happy thanksgiving.

---

### Official Review · Reviewer_BuXH · 2023-10-29

**Soundness:** 3 good
**Presentation:** 2 fair
**Contribution:** 2 fair
**Rating:** 5
**Confidence:** 3

**Summary:**

This paper proposes differentially private versions of Min-wise Hashing One Permutation Hashing. By restricting the hash values to b-bit integers, the proposed algorithms achieve differential privacy by applying the randomized response technique over the finite space of output values. The numerical performance of proposed algorithms is validated on real data sets.

**Strengths:**

* The proposed algorithms are intuitive and easy to understand/implement.
* The author(s) took care to define their notion of neighboring data sets and the implied privacy guarantees ("attribute-level").
* There is an extensive set of numerical experiments on real data sets.
* The concise introduction to Min-wise Hashing and One Permutation Hashing allow readers who are unfamiliar with the field to quickly pick up the necessary background and understand the problems studied by this paper.

**Weaknesses:**

* Lack of utility analysis. As motivated in the introduction, hashing is frequently for the purpose of approximately calculating the similarity between two high-dimensional binary vectors. The paper also mentions the bias and variance of non-private estimators of the Jaccard similarity. However, except in Section A of the Appendix, there is not much mention of how the estimation accuracy, using the proposed hashing methods, depends on the privacy parameters, dimension of data, number of bins, etc. The discussion in Section A is also limited in two ways: (1) the variance analysis is only empirical; (2) it only considers replacing the non-private hash values in a specific estimator defined in equation (2) by private hash values, while there is no clear justification given for why this particular estimator is still the best candidate for private estimation of Jaccard similarity. Some utility analysis can be very useful for assessing the proposed algorithms in their own right, or relative to existing methods. For example, one advantage of this paper's set up, as suggested at the end of Section 1.2, is that it makes no assumptions about whether hash functions are kept private. However, without understanding the cost of this generality, potential users may still have difficulty in choosing between competing methods of DP hashing.

* Some gap in motivating One Permutation Hashing and b-bit coding. The paper mainly studies One Permutation Hashing with hash values restricted to b-bit integers, but the justifications for these choices appear somewhat ambiguous, especially to readers without much background in this topic. For example, it would be helpful if the last sentence of the first paragraph in Section 3.1 can include some more details (why unstable? violating the triangle inequality with respect to which metric?). For another example, restricting hash values to b-bit integers appears to be crucial to the DP algorithms (otherwise, the randomized response mechanism may not apply), however the paragraph on b-bit coding on page 3 feels rushed and leaves out some important details (without considering privacy yet, is there any accuracy loss by restricting to b-bits? How is b usually chosen? Is b-bit coding convenient, or rather, necessary for DP hashing?)

**Questions:**

In addition to the questions mentioned in "weaknesses":

* On the usefulness of "attribute-level" privacy, is there any application of hashing/Jaccard similarity where it is more natural to consider the entire D-dimensional vector, as opposed to one coordinate of the vector, as the data of "one individual"?

* In the DP literature, it is customary to set $\delta$ to be smaller than 1 divided by the data set's size ($1/D$ in the case of this paper). What value of $\delta$ do you recommend in general, and what is its relationship to $1/D$?

* Does your approach require that the dimension $D$ is fixed and/or public information? Strictly speaking, the notion of DP also allows adding/deleting one record from the data set, as opposed to merely replacing one record and keeping the data size unchanged.

---

> ### Author Response · Authors · 2023-11-18
> **Thank you**
>
> Dear Reviewer BuXH:
>
> Thanks for your feedback on our submission.
>
> W1. As you kindly pointed out, in principle the variance of the (unbiased) Jaccard estimator from DP-OPH can be computed. However, given the already very complicated combinatorial form of the variance of non-private OPH, the expression will be extremely sophisticated also considering the b-bit coding and DP flipping. Thus, a straightforward analytical comparison is hard. We hope the empirical MSE comparisons in the appendix suffice to demonstrate that DP-OPH-re has smaller variance. Moreover, we know that in general the extra variation brought by DP would be largely determined by $N$, the privacy discount factor in DP-OPH-re and DP-OPH-fix. In Figure 1, we compare the $N$ of difference methods to intuitively show the advantage of the proposed approach in terms of privacy. These comparisons, together with the extensive experiments, show that the proposed algorithms provide better utility.
>
> W2. Clarification on the background: Thanks for the nice suggestion. When we drafted the paper, we had to omit some contents due to the space limitation. We will definitely add more clarifications on the background of OPH and b-bit coding strategy. 1) Instability of naive OPH: without densification, the Jaccard estimation of OPH is (# of hash collisions)/(# of jointly non-empty bins). For sparse data, the denominator could be very small. Thus, the estimated value may vary a lot. For the metric, we refer to the Hamming distance. 2) We will add more references on b-bit coding, especially the empirical evaluations. Using b-bit coding will lead to some performance drop, which increases as $b$ gets smaller. However, truncating the hash values in this way is a standard step for the convenient use in practice. Since the vanilla OPH or MH hash values are large integers, b-bit coding is important for our DP methods---it limits the output domain so that bit flipping is able to guarantees a certain level of utility.
>
> Q1. As we mentioned/cited in the paper, attribute-level DP is a standard setup in the literature of private sketching/hashing, e.g., [Kenthapadi et al, 2013], [Stausholm, 2021], [Zhao et al, 2022], [Smith et al, 2020], [Dickens et al, 2022]. The reason is that, our goal is to privatize the output hash values, and they are representations (or called "signatures" in some fields) of each data vector. It then becomes natural to treat a data vector as the "database", and define "neighboring" on the coordinate level.
>
> We may also "consider the entire D-dimensional vector" as you mentioned. We discuss two settings here. First, for real data vectors and when the randomness of the hashing algorithm is internal and private [Blocki et al.2012], there is a privacy definition that $\|u-u'\|\leq 1$ for neighboring vectors $u$ and $u'$. For binary data, this is equivalent to our definition. We may extend the definition to "differing in multiple coordinates", which is similar to the notion of "group privacy". This requires more involved analysis, which could be an independent study for future investigation. In this paper, we follow the common definition of "one coordinate change" in most related works on private sketching.
>
> Second, if what the reviewer suggested is to protect arbitrarily different vectors $u$ and $u'$, then the problem becomes impossible to solve because the output hash values of two arbitrary vectors $u$ and $u'$ in $R^d$ can also be arbitrarily different. It is hard to find useful data representation while requiring that the representation of any data vector should look similar (which is required by DP).
>
>
> Q2. Thanks for the question and we would like to provide our thoughts here. In some DP papers, $\delta=1/n$ ($n$ is the size of the database) is used, but in our understanding this might be largely because the interpretation "at most one user's record will be leaked" seems more consistent with the "unit change" notions in the definition of DP. There are also many other papers that simply choose a deterministic value for $\delta$, which, based on our experience, is usually between $\delta=10^{-6}\sim 10^{-4}$. In our work, we presented the results with $\delta=10^{-6}$ which is smaller than $1/D$ for all the datasets in our experiments. The comparisons indeed hold for other $\delta$ values that we have experimented.
>
> Q3. $D$ can be publicly known and fixed. When we interpret binary vectors as sets, $D$ is the size of the universe of items, and "1" means the item is in the set. So basically, in our problem we assume the universe itself does not change. The add/deletion operation is on the set level---the data coordinate of $x$ changing from 0 to 1 means "adding the item to the set", and changing from 1 to 0 means "deleting the item from the set".
>
> We hope our rebuttal adequately answers your questions. Please kindly let us know if there are any further questions. Thank you.

---

### Official Review · Reviewer_ZyNs · 2023-11-01

**Soundness:** 3 good
**Presentation:** 3 good
**Contribution:** 2 fair
**Rating:** 3
**Confidence:** 3

**Summary:**

The Jaccard similarity is a measure that allows to compute the amount of items that two parties have in common. It is widely used to compute the similarities of two entities or individuals (e.g. preferences, genomes). Since its exact computation is expensive, it is commonly estimated via sketches such as Min-Hash (MH) and One Permutation Hash (OPH). An interesting problem is to provide privacy preserving estimations in order to hide sensitive information when the data of individuals is involved.

The proposed work addresses the privacy preserving computation of Jaccard estimations. It proposes privacy preserving versions of different variations of OPH and MH, proves their differentially private guarantees and compares their accuracy under different parameters.

**Strengths:**

I think that obtaining privacy preserving Jaccard distance estimations is an interesting problem.

The paper has properly addressed its claims by providing proofs of privacy and extensively evaluating their results.

Key aspects of the contribution have been in general well presented.

**Weaknesses:**

However, I found several problems that I list below.

1- Impact of the results: all algorithms are proven private, but their privacy-accuracy tradeoffs do not seem widely applicable in practice. The evaluation of Section 4 shows that estimations are accurate for only very large values of $\epsilon$ (i.e. between 10 and 50) and therefore a large privacy loss. A partial exception is DP-OPH-rand, which shows fairly better accuracy already when $\epsilon$ reaches 5. However, I its behavior is not clear since plots focus on a larger scale and we cannot see in detail what is happening in the range [0.1, 10].

2- Novelty: I do not find substantial novelty in the work. The application of randomized response to MH sketches is already present in Aumüller et al. (2020). It is true that the authors found a mistake on the proof of this related work and corrected it, but as said in the paper, this mistake is minor. It seems that the application of these ideas to the proposed OPH variations do not have substantial changes. If the authors could clearly explain the novelty in the content of their proofs with respect to Aumüller et al. (2020), I will take this point back and re-asses my score.

3- Clarity: Even if I think the paper is overall well presented, it still requires more clarity in the technical sections.

3a- The degree of "locality" of the DP guarantee could be more clearly explained. In the contribution, there is no need to trust a central party to take two vectors and output a noisy estimation of its Jaccard distance as it would be needed in central DP. Clarifying this aspect would also help to better motivate the scope of application of the result.

3b- The results presented in Section 3.2 of the paper are not always self contained and could be better presented.

**Questions:**

Please comment on points 1 and 2 raised in as weaknesses.

---

> ### Author Response · Authors · 2023-11-18
> **Thank you**
>
> Dear Reviewer ZyNs,
>
> Thanks for your feedback on our work.
>
> 1. DP has been deployed in a wide range of fields and applications, and the feasible $\epsilon$ depends on the specific use case. For example, in the tutorial [How to dp-fy ml: A
> practical guide to machine learning with differential privacy, Ponomareva et al., 2023, Section 5.2] by Google, the researchers defined $\epsilon<10$ as "reasonable privacy protection". For the U.S Census use case [The use of differential privacy for census data and its
> impact on redistricting: The case of the 2020 U.S. Census, Kenny et al., 2021], $\epsilon=12.2$ is chosen. The PyTorch tutorial of DP-SGD training sets $\epsilon=47.2$ (https://github.com/pytorch/opacus/blob/main/tutorials/building_image_classifier.ipynb). Therefore, depending on the application and privacy requirement, we believe that different $\epsilon$ values in DP-OPH are useful in difference use cases.
>
> In our plots, we presented $\epsilon\in [1,50]$. We referred to the recent paper [Zhao et al., Differentially private linear sketches: Efficient implementations and applications. NeurIPS 2022] when presenting the range of $\epsilon$. [Zhao et al. 2022] studied DP count sketch, which is similar to MinHash/OPH in terms of DP since count sketch also produces multiple hash values for each data vector. The range of $\epsilon$ used in [Zhao et al. 2022] is $[2.45, 33.5]$ (converted from zCDP). We presented larger $\epsilon$ to demonstrate the difference between DP-OPH-rand and densified DP-OPH-re.
>
> For our proposed methods, DP-OPH-rand typically performs well with $\epsilon<10$. The densified DP-OPH-re may outperform DP-OPH-rand when $\epsilon>5\sim 15$, depending on the dataset. Therefore, we hope that all the variants could find useful use cases in practice.
>
> 2. The analysis of DP-OPH is much more involving than DP-MH with some complicated combinatorial calculations due to the binning operations in OPH. Rigorously obtaining the flipping probability requires non-trivial efforts. On the high level, we propose DP-OPH and show that it can significantly improved DP-MH. We believe these new algorithm designs and results themselves are novel contributions to the literature of private sketching/hashing.
>
> 3. Thanks for the nice suggestions. Not requiring a trusted central server is indeed one advantage of our proposed methods. We will highlight it in the revision.
>
> Again, we appreciate your valuable feedback. Please kindly let us know if there are any more questions that we can help clarify. Thank you.

---

> ### Comment · Reviewer_ZyNs · 2023-11-21
>
> I thank the authors for their answers. I further comment below referencing points 1 and 2 of the rebuttal.
>
>
> 1- The authors cited [1,2,3,4] as evidence that $\epsilon \in [10, 50]$ provide reasonable privacy. I will argue why I am not convinced by these arguments:
>
>
> 1a- I would first like to point out that [1] is a guide that mainly focuses on the privacy of broader ML tasks. The objective of the current submission is privatizing a low level primitive that, in machine learning practice, will be part of larger protocols. The current submission provides privacy (with reasonable accuracy) only for upper half Tier 2 $\epsilon$ (i.e. $5 <  \epsilon <= 10$) and Tier 3 $\epsilon (10 <  \epsilon)$. In  Section 5.2 of [1], it is argued that Tier 3 $\epsilon$ is considered almost as no privacy and Tier 2 $\epsilon$ is considered reasonably private *only under certain conditions*. The guide specifies that the choice of what is the privacy unit plays an important role. In this contribution, the privacy unit is not even an individual but an item of the individual. The guide also remarks that Tier 2 $\epsilon$ is sometimes considered private because ML privacy accounting techniques assume strong privacy leakages. However, these leakages do not occur often in practice (e.g. the leakage of intermediate iterations). Importantly, it is highlighted in [1] that
>
> *"from the DP point of view, the ε ∼ 10 guarantees might seem dubious" as "this value of ε would translate into the probability of a particular outcome changing by 22026 times on two datasets that differ only by one instance (in case of instance level privacy)."*
>
>
>
> 1b- Is is highlighted in page 2 of [2] that $\epsilon$=12.2 *"represents a relatively high privacy loss budget"* and that even $\epsilon=4$ is high. In particular it is highlighted that *"there may not be an overlap between the values of $\epsilon$ that are considered stringent enough for privacy purposes" and $\epsilon \in [4,12]$*.
>
>
>
> 1c- [3] is a tutorial to understand basic concepts about DP ML and therefore it cannot serve as a reference of privacy in real applications
>
>
>
> 1d- The authors say that they take [4] as a reference for the (translated from zCDP by the authors) range [2.45, 33.5] of $\epsilon$. I do not see anywhere in [4] where it is argued that such range could be considered as good privacy parameters in sketching. Moreover, I think the range of parameters in [4] is different than what is claimed by the authors. Their experiments are done with $\rho \in {0.1, 1, 10 }$ of zero concentrated DP. I used Lemma 2.9 of [4] to convert zCDP to DP, and these values of $\rho$ would fall in the range [1.33, 10.79] for $\delta=10^{-6}$ as in you use in your experiments. Could you clarify why these values are different?
>
> Given the points above, I am strongly discouraged to think that your technique provides reasonable levels of privacy
>
> Refs.
> [1] How to dp-fy ml: A practical guide to machine learning with differential privacy, Ponomareva et al., 2023
>
> [2] The use of differential privacy for census data and its impact on redistricting: The case of the 2020 U.S. Census, Kenny et al., 2021
>
> [3] The PyTorch tutorial of DP-SGD training sets (https://github.com/pytorch/opacus/blob/main/tutorials/building_image_classifier.ipynb).
>
> [4] Zhao et al., Differentially private linear sketches: Efficient implementations and applications. NeurIPS 2022
>
>
>
>
> 2- I am now more convinced that your proposed techniques contain additional novel content with respect to Aumüller et al. (2020). However, I am not sure. After a more detailed verification I will raise my score in the affirmative case. However, given point 1 above I doubt that I would positively support acceptance.

---

> > ### Author Response · Authors · 2023-11-22
> > **Reply to the additional comments**
> >
> > Dear Reviewer ZyNs,
> >
> > Thanks for the further comments and discussion. We are glad that our novel contributions are more clear now. We would like to provide our thoughts as below.
> >
> > About the $\epsilon$ calculation: For the $\epsilon$ range in [Zhao et al. 2022], their Lemma 2.9 says that $\rho$-zCDP can be translated into $(\rho+2\sqrt{\rho\log(1/\delta)},\delta)$-DP. Note that $\log(1/\delta)\approx 13.82$ when $\delta=10^{-6}$. So $\rho=0.1$ gives $0.1+2\sqrt{0.1\times 13.82}=2.45$, and $\rho=10$ gives $10+2\sqrt{10\times 13.82}=33.5$.
> >
> > We would like to point out that, "privatizing low level primitive" as DP-OPH is in general harder than privatizing the "output of an algorithm" in many cases. DP-OPH casts privacy on the data level---it produces data signatures for efficient large-scale search and learning, for each data vector. This allows private data sharing where we can simply share the transformed dataset with others for any subsequent task, and the task would also preserve DP. This is not available if we only privatize the output of a specific algorithm. In our "attack model", the adversary can access the DP-OPH representation of each data vector and the hash functions used. To a large extent, this is conceptually the same as privatizing the raw data vectors, which is harder than, for example, privatizing the algorithm of computing the samples statistics (e.g., mean, median) of many vectors.
> >
> > Also, as mentioned in the paper, the case where hash functions of OPH are publicly known is more difficult. In fact, it is not hard to show that, if the hash functions are assumed private, OPH and MinHash themselves possess good DP properties when the data is not very sparse and would require much less DP perturbation. However, this simpler setup is less practical, so we studied the known hash function case. These are the resons that the $\epsilon$ value in our paper and [Zhao et al. 2022] (which has a similar DP-sketching setting) needs to be a bit larger ($5\sim 15$) to achieve good utility.
> >
> > In the PyTorch official tutorial on DP-SGD, we see that even with $\epsilon=47$, the resnet accuracy is still way lower than the non-private model. While this is not a formal publication, we think it might still be a good example that the $\epsilon$ value should really depend on different privacy settings and use cases.
> >
> > Again, we sincerely thank the reviewer for the feedback and discussion. We hope our reply helps the reviewer better understand/appreciate the privacy setting and application scenarios of our work. Thank you and happy thanksgiving.

---

> > > ### Comment · Reviewer_ZyNs · 2023-11-22
> > >
> > > Dear authors,
> > >
> > > I apologize for the miscalculation! Indeed I agree with your translation from zCDP to DP. I will take that into account in my final score. However, given my previous points 1 and 2, it is likely that it will remain below the acceptance threshold. It is not the difficulty in obtaining privacy what is my main concern of point 1, but the lack of meaningful privacy guarantees and, therefore, the impact of the result.
> > >
> > > Happy thanks giving to you too.

---

### Official Review · Reviewer_mGib · 2023-11-05

**Soundness:** 2 fair
**Presentation:** 3 good
**Contribution:** 2 fair
**Rating:** 5
**Confidence:** 3

**Summary:**

This manuscript proposes DP-One Permutation Hashing, which (the hashing itself) is an improved approach of the minwise hashing that is used for Jaccard similarity estimation. The DP-OPH framework has three variants based on different densification approaches. And the main technique is the randomized response tailored to the hash values. The analysis is given for the approximate-DP guarantee. To establish a comparison, DP-Minwise Hashing is also discussed, however experimental results show that DP-OPH outperforms DP-MH very frequently.

**Strengths:**

1. In general, the paper is well-written with clarity. The related work is carefully discussed.
2. Recently there are more increasing interest in studying DP for hashing, sketching, etc. This work contributes to the developing topic.
3. Algorithms for three densification approaches are discussed, and this work further fixed a small error of previous work. The correctness should be OK. -- I checked 60% of the proof and skip the rest due to the timeline. But I am happy to come back to it should there be any concern.

**Weaknesses:**

1. My major concern is the missing analysis of the utility. From what I read, the DP guarantee is shown but the utility is justified by experiments.  It is somewhat an incomplete work if no upper/lower bound on the utility is discussed, and further weaken the some of the experimental results. For example: Though the proposed DP-Minhash and DP-OPH are similar at a schematic level, it is unfair if the DP-MH is not the optimal algorithm however DP-OPH is. I will say more in the questions section.

2. The technique seems to be randomized response mechanism -- it is quite naive.

3. There should be discussion on pure-DP.

**Questions:**

1. What is $J$ in the equation after equation 2 on page 3?
2. In the section of __Privacy statement and applications__, it is mentioned the privacy model is "attribute-level". Are there other privacy models for this type of problem? (Such as node/edge level privacy for graphs)
3. Corresponding to the major concern. I briefly checked two previous papers on this topic and really appreciate it if the authors help me understand why the analysis of utility is not available in this work. In [2], it is claimed that One Permutation Hashing outperforms Minwise Hashing, and the theoretical analysis includes two lemma on page 5: Lemma 1 says the expected number of 'jointly empty bins' has an upper bound; Lemma 2 says the estimator of the resemblance is unbiased. Would you please explain intuitively how these two analysis show the better utility?
4. To continue, in [1], where there is a small error of probability distribution fixed in this paper, the authors gave utility analysis. Looking at lemma 5 and theorem 2 on page 6 (arxiv version), the upper bound of additive error seems to be $O(\frac{1}{\sqrt{n\varepsilon}})$ by taking $\alpha =1, \tau = n$ if I did not misunderstand. Why cannot we do a similar analysis on the algorithms proposed here?

Minor: Maybe explicitly state the randomized response mechanism in the preliminary.

[1] Differentially Private Sketches for Jaccard Similarity Estimation.
[2] One Permutation Hashing

---

> ### Author Response · Authors · 2023-11-18
> **Thank you**
>
> Dear Reviewer mGib:
>
> Thanks for your feedback on our submission.
>
> Q1. $J$ is the Jaccard similarity between $u$ and $v$ defined in equation (1), i.e., $J = J(u,v)$. We omitted "$(u,v)$" here. We will make the notation more clear. Thanks for pointing it out.
>
> Q2. As we mentioned/cited in the paper, attribute-level DP is a standard setup in the literature of private sketching/hashing, e.g., [Kenthapadi et al, 2013], [Stausholm, 2021], [Zhao et al, 2022], [Smith et al, 2020], [Dickens et al, 2022]. The reason is that, our goal is to privatize the output hash values, and they are representations (or called "signatures" in some fields) of each data vector. Thus, it is natural to treat a data vector as the "database", and define "neighboring" on the coordinate level.  For real data vectors and when the randomness of the hashing algorithm is internal and private, there is another privacy definition that $\|u-u'\|\leq 1$ for neighboring vectors $u$ and $u'$. However, for binary data, this is equivalent to our definition.
>
> W3. The DP-OPH-rand variant (Algorithm 5) with random bits for empty bins achieves $\epsilon$-DP, see Theorem 3.3.
>
> **Regarding your major concern**
>
> Q3. About the OPH paper: We appreciate your very detailed research on the prior work. Actually, the Arxiv version of the NIPS 2012 paper https://arxiv.org/pdf/1208.1259.pdf included more details. The exact analysis of the OPH type algorithms is highly complicated with involved combinatorial calculations. In Lemma 6 and Lemma 7, the authors showed that the approximate Jaccard estimation variance ratio of OPH over minhash is smaller than 1. In Figure 8, we see that empirically OPH can perform better than minhash with smaller MSE.
>
> Q4. The error bound in [1] is a simple consequence of applying Chernoff bound on the average of iid Bernoulli random variables. A minor point: in their Theorem 2, $\tau$ does not equal to $n$. Actually, it is the number of non-zeros in the data vector which can be much smaller than $n$. So the rate is $O(1/\sqrt{\tau \epsilon})$.
>
> For DP-OPH, the analysis would be much harder and challenging. The key reason is that the hash values of OPH are not independent---recall that OPH is a "sample-without-replacement" approach, so the hash values are correlated to each other. Thus, the standard concentration tools would not be applicable here. Moreover, even if we compute the variance explicitly, it would be extremely sophisticated considering both the $b$-bit coding and DP flipping. Thus, a direct analytical comparison would be difficult. Given these reasons, we chose to present the comparison on the privacy discount factor $N$ in Figure 1, an empirical comparison of the Jaccard estimation variances in Appendix A, and justify the performance of DP-OPH methods through extensive experiments on datasets from computer vision, biology, and the web. We hope these results are convincing to show the effectiveness of DP-OPH methods.
>
> We hope our reply clarifies your questions. Should there be any further questions, please kindly let us know. Thanks again for your efforts in reviewing our paper.

---

### Official Review · Reviewer_QDZh · 2023-11-06

**Soundness:** 3 good
**Presentation:** 3 good
**Contribution:** 2 fair
**Rating:** 5
**Confidence:** 2

**Summary:**

This paper considers making a version of min-wise hashing algorithm differentially private.

Minwise-hashing is a well know algorithm for estimating Jaccard Similarity between large sets. Jaccard similarity is a widely used similarity measure in many situations including document similarity. In the standard scheme, to improve quality, multiple random permutations need to be used to get multiple hash values. One Permutation Hashing (OPH) is a variant that uses just one random permutation to generate multiple hash values and hence reduces complexity.

The main contribution of this paper is making OPH differentially private (in the Dwork et al DP framework).   They give DP OPH for three versions of OPH what the authors call OPH with fixed densification and OPH with re-randomized densification (densification is a technique to deal with "empty bin" problem that arise in OPH), and OPH with Random bits. The authors also experimentlaly evaluate their   algorithms.

**Strengths:**

The main strength is that the paper considers DP version of a very practical algorithm and hence could be of use where privacy is important. The paper also experimentally evaluate their algorithm.

**Weaknesses:**

The methods seem to be straight forward and hence scientific content appears not that substantial.

**Questions:**

I do not have any questions at this point.

**Details Of Ethics Concerns:**

No Concern.

---

> ### Author Response · Authors · 2023-11-18
> **Thank you**
>
> Dear Reviewer QDZh:
>
> Thanks for your feedback.
>
> Private sketching/hashing methods have attracted more and more attention in the research community recently. However, min-wise hashing and its efficient alternative, one permutation hashing, have not been studied under the differential privacy model. As we have extensively cited in the paper, these important algorithms have many applications in practice, so we believe providing private versions of these methods would be helpful to many researchers and practitioners. We actually tried several different ways to achieve DP, and it turns out that the "classic" generalized randomized response technique is the most effective one and well fits the discrete nature of our problem. While the DP tool is not completely new, the main technical contributions of our work is to properly specify the privacy parameters in DP-OPH (Lemma 3.1), which requires non-trivial design and analysis. We also provided three variants that can be chosen flexibly under different scenarios. We hope that our work can serve as a good guidance and reference for private hashing methods of the Jaccard similarity.
>
> We hope our rebuttal could help the reviewer better appreciate our contributions. Again, thanks for your review of our work.

---

> > ### Comment · Reviewer_QDZh · 2023-12-04
> > **Response**
> >
> > I thank the authors for the response. I will retain my review.

---

### Author Response · Authors · 2023-11-21
**Any further questions?**

Dear Reviewers,

Again, we sincerely appreciate your efforts in reviewing our work and providing valuable feedback. We hope our rebuttal addresses the raised questions. Please kindly let us know if there are further questions we can help clarify before the author-reviewer intersection phase ends. Thank you very much.

Best,
Authors

---

### Meta-Review · Area_Chair_4qqV · 2023-12-06

**Metareview:**

This paper studies the question of differentially private (DP) sketching for Jaccard similarity. In the non-private setting, the classical sketch is the so-called MinHash where we take a permutation over the domain and then let the sketch be the minimum value in the image of the set we want to hash. By looking whether this minimum is the same, we can derive an (unbiased) estimator for Jaccard similarity. The variance of the estimate can be reduced further by taking multiple (say $k$) permutations. However, this can be expensive so a slight adaptation called One-permutation hashing (OPH) has been proposed (Li et al., 2012). Instead of using multiple permutation, OPH uses a single permutation but dividing the domain into $k$ buckets of equal size and record the minimum in each bucket instead. While this is effective computationally, it can be bad for small sets and most buckets will be empty. This led to "densification" tricks (Shrivastava et al., 2017; Li et al., 2019) that, roughly speaking, fills empty buckets using some other random non-empty bucket.

The main contribution of the paper is to make the OPH sketch and its densification variants private. The idea is simple: just apply randomized response to each bucket. The privacy of vanilla OPH is obvious. The main technical contribution of this work is to analyze privacy of the densification variants, which are non-trivial since the copying can significantly increase the sensitivity. The authors then provide experimental results on real world datasets.

## Strengths

- Jaccard similarity is a widely used measure and DP sketches for Jaccard similarity can thus have an important impact.

## Weaknesses

- **Larger than ideal privacy parameters:** The $\\epsilon$ values required for the sketches to provide meaningful results is quite large ($> 5$). Although this is sometimes seen as acceptable in ML or large-scale statistic tasks, having such a large $\\epsilon$ for a basic task like nearest neighbor search (which is often used as a subroutine of downstream tasks) does not seem practical.

- **Technical novelty**: Among the different methods tested, it turns out that the most basic DP-OPH-rand beats the other methods in meaningful regime of $\\epsilon$. This DP-OPH-rand algorithm simply uses the naive randomized response mechanism with the standard analysis. Thus, the technical contribution is less clear (given that the majority of the paper is spent analyzing the privacy of the other methods).

- **No formal theoretical utility analysis:** Although the paper provides a number of experimental evaluations, no formal utility bound is given. This makes it quite hard to understand the behavior of the sketches w.r.t. different parameters (and thus hard to choose these parameters effectively). At least some crude (e.g. asymptotic) utility bounds should be provided.

**Justification For Why Not Higher Score:**

The best performing algorithm in experiments DP-OPH is the trivial adaptation of OPH in the DP setting, so it is not entirely clear to me what is the point of this paper. Furthermore, all algorithms presented require very large $\\epsilon$'s and are simply impractical.

**Justification For Why Not Lower Score:**

N/A

---

### Decision · Program_Chairs · 2024-01-16

Reject